# Delivery of a BET protein degrader via a CEACAM6-targeted antibody–drug conjugate inhibits tumour growth in pancreatic cancer models

Youya Nakazawa [1,3] ✉, Masayuki Miyano[1,3], Shuntaro Tsukamoto[1], Hiroyuki Kogai [1], Akihiko Yamamoto[1], Kentaro Iso [1], Satoshi Inoue[1], Yoshinobu Yamane[1], Yuki Yabe[1], Hirotatsu Umihara[1], Junichi Taguchi[1], Tsuyoshi Akagi[1,2], Atsumi Yamaguchi[1], Minaho Koga[1], Kohta Toshimitsu[1], Toshifumi Hirayama[2], Yohei Mukai [2] & Akihito Machinaga[1,2]

Pancreatic ductal adenocarcinoma (PDAC) has the worst prognosis of all cancers. To improve PDAC therapy, we establish screening systems based on organoid and co-culture technologies and find a payload of antibody–drug conjugate (ADC), a bromodomain and extra-terminal (BET) protein degrader named EBET. We select CEACAM6/CD66c as an ADC target and developed an antibody, #84.7, with minimal reactivity to CEACAM6-expressing normal cells. EBET-conjugated #84.7 (84-EBET) has lethal effects on various PDAC organoids and bystander efficacy on CEACAM6-negative PDAC cells and cancer-associated fibroblasts. In mouse studies, a single injection of 84-EBET induces marked tumor regression in various PDAC-patient-derived xenografts, with a decrease in the inflammatory phenotype of stromal cells and without significant body weight loss. Combination with standard chemotherapy or PD-1 antibody induces more profound and sustained regression without toxicity enhancement. Our preclinical evidence demonstrates potential efficacy by delivering BET protein degrader to PDAC and its microenvironment via CEACAM6-targeted ADC.

Pancreatic ductal adenocarcinoma (PDAC), a common type of pancreatic cancer, is usually diagnosed at advanced stages (stage III, about 20%; stage IV, about 50%). Most treatment options are ineffective, resulting in 10% survival for all stages and 3% survival for stage IV at 5 years in the United States[1]. Predictions of cancer incidence and deaths suggests that deaths from pancreatic cancer will increase continuously, and pancreatic cancer will become the second leading cause of cancer-related death in the US by 2030[2,3]. One of the promising classes of highly potent modalities is antibody–drug conjugate (ADC)[4], in which an antibody is conjugated to a small compound known as the payload. The combination of high selectivity and stability of the antibodies with high cell-killing activity of the payload enables specific killing of malignant cancer cells. Another key element of ADCs is the "bystander effect," namely the effect of diffused payloads, after targeted delivery, on surrounding cancer or stromal cells lacking target expression[5]. This effect might be crucial to the fight against PDAC, because tumor heterogeneity has an impact on the prognosis[6].

Despite the increasing role of chemotherapy in pancreatic cancer treatment, current options are very limited. For neoadjuvant chemotherapy and first- and second-line chemotherapy, gemcitabine

[1]Tsukuba Research Laboratory, Eisai Co., Ltd., Ibaraki, Japan. [2]KAN Research Institute, Inc., Kobe, Japan. [3]These authors contributed equally: Youya Nakazawa, Masayuki Miyano. ✉e-mail: youya.nakazawa1216@gmail.com

(GEM) + nanoparticle albumin-bound paclitaxel (nab-PTX), and FOL-FIRINOX or modified FOLFIRINOX regimens are recommended in National Comprehensive Cancer Network Guidelines (NCCN Guidelines®). For adjuvant chemotherapy, S-1 (only in Japan), GEM + capecitabine, and modified FOLFIRINOX regimens are recommended. Cisplatin followed by olaparib, larotrectinib or entrectinib, and pembrolizumab are recommended for BRCA1/2- or PALB2-mutated cancers, NTRK gene-fusion-positive cancers, and MSI-H/dMMR cancers respectively. Liposomal irinotecan is recommended as a second line after GEM-based therapy. For later lines, best supportive care is recommended. Therefore, there are largely only three options for most PDAC patients: GEM-based, 5FU-based, or irinotecan (SN38)-based chemotherapy.

Over the past decade, several bromodomain and extra-terminal (BET) inhibitors have entered clinical trials against various solid tumors and haematological malignancies. However, the efficacy of these inhibitors has been very limited due to severe drug-related toxicities[7]. For clinical application of BET modulators, a specific delivery system to tumors would be necessary.

In this study, we identify BET protein degrader[8,9] EBET as a payload candidate for PDAC. As a target for the ADC, we select carcinoembryonic antigen-related cell adhesion molecule 6 (CEACAM6, also known as CD66c) and develop an antibody, #84.7. EBET-conjugated #84.7 (84-EBET) induces marked tumor regression in various PDAC-patient-derived xenograft (PDX) models and has a combined effect with standard chemotherapy or programmed death 1 (PD-1) antibody without substantial toxicity. These findings might facilitate development of a superior PDAC therapy and provide hope to patients.

## Results

### Organoid culture phenotypically mimics the original PDAC-PDX tumor
PDAC is one of the most chemoresistant cancers. This chemoresistance cannot be fully evaluated in conventional 2D cultures of PDAC cell lines, because there are many factors related to chemoresistance in human PDAC, including heterogeneity within cancer cells or stromal cells, a dense fibrotic stroma (causing fibrosis or desmoplasia), an immunosuppressive microenvironment, and complex interactions between cancer cells and stromal cells[6]. To address these complexities, we tried to establish an organoid system co-cultured with stromal cells from PDX tumors.

To select primary PDAC models, we classified our proprietary PDAC-PDX panel into known prognostic molecular subtypes, namely classical tumor vs. basal-like tumor[10], and Myc-signal low vs. Myc-signal high[11], by using RNA sequencing (RNA-seq) data on PDX tumors (Supplementary Fig. 1a). We then chose two contrasting PDAC-PDX models. PC-3 (not a prostate cancer model) is a model with a molecular subtype of classical tumor and Myc-signal low which are associated with better prognosis in PDAC patients, whereas PC-42 is a model with a molecular subtype of basal-like tumor and Myc-signal high which are associated with poorer prognosis in PDAC patients. Mouse studies revealed that the PC-3 tumor had a moderately differentiated and GEM-sensitive phenotype with tumor shrinkage, whereas the PC-42 tumor had a poorly differentiated and GEM-resistant phenotype without tumor shrinkage (Supplementary Fig. 1b, c).

Organoids were established directly from digested fragments of PDX tumors containing both cancer cells and stromal cells. Organoids from the PC-3 and PC-42 tumors reflected the original tumor phenotypes in terms of GEM-sensitivity and differentiation (Supplementary Fig. 1d, e). For more detail, we performed an immunofluorescence analysis of PDX tumors and organoids. BC2LCN (N-terminal domain of the lectin BC2L-C from *Burkholderia cenocepacia*) has been reported by our former collaborators to mark human undifferentiated pluripotent cells[12], cancer stem-like cells[13], and some pancreatic cancer cells[14,15]. Almost all the E-cadherin+ cancer cells were BC2LCN negative

in PC-3 tumor, whereas E-cadherin+ cancer cells were BC2LCN positive in PC-42 tumor (Supplementary Fig. 2a). After GEM treatment, the abundance of BC2LCN+ cancer cells increased in the PC-3 tumor but remained unchanged in the PC-42 tumor. The surviving BC2LCN+ cancer cells were Ki67+ cycling cells. In both models, the tumors were highly fibrotic and collagen I rich. Similarly, the PC-3 organoid showed an increase in abundance of BC2LCN+ cells and collagen I staining after GEM treatment (Supplementary Fig. 2b). The PC-42 organoid showed greater BC2LCN+ abundance and collagen I staining with or without GEM treatment compared with the PC-3 organoid. Taken together, these findings indicated that our organoid culture system would be a good surrogate for in vivo PDAC tumors to help us understand the dynamics of PDAC cells and stromal cells.

### Compound screening with organoids identifies a compound, EBET-1055, effective against PDAC
By using the organoid system, we screened our proprietary compound library containing known reference compounds (Supplementary Fig. 3a) and original compounds to select a payload. We newly identified the BET protein degrader EBET-1055 as a compound that was highly potent in both PC-3 and PC-42 organoids (Fig. 1a and Supplementary Fig. 3b). Importantly, PC-3 organoids were sensitive to standard chemotherapy drugs, whereas PC-42 organoids were resistant to them (Fig. 1a). The most commonly used payloads, maytansine and monomethyl auristatin E (MMAE), could not kill all the PC-3 organoids and were completely ineffective in the PC-42 organoids. Some DNA-binding payloads, namely PNU-159682[16], calicheamicin[17], and SJG-136 (a derivative of pyrrolobenzodiazepine dimer)[18], were effective against PC-42 organoids, but the half maximum inhibitory concentration ($IC_{50}$) was about 10 to 100 nM, much higher than the $IC_{50}$ values reported in 2D culture[16–18].

Subsequently, we screened the hit compounds with a stromal signal reporter assay. As stromal cells, we tested human pancreas and liver stellate cells (PSCs and LSCs) and chose LSCs with high reporter activity for screening. In this screening, we examined the effects of compound treatment on interferon-stimulated response element (ISRE, for the interleukin (IL)-1 signal), nuclear factor kappa-light-chain-enhancer of activated B cells (NF-κB, for the IL-1 signal), signal transducer and activator of transcription 3 (STAT3, for the IL-6 signal), and small mothers against decapentaplegic 2/3 (SMAD2/3, for the TGF-β signal) reporter activity on LSCs in co-culture with cancer cells isolated from PC-3 tumors. These stromal signals are important in the function of cancer-associated fibroblasts (CAFs) in PDAC[19–22]. All reporter activities on LSCs were enhanced by co-culture with PC-3 cancer cells according to the basal values relative to monoculture (ISRE: 32.8, NF-κB: 2.4, STAT3: 2.0, SMAD: 9.5; Fig. 1b). Among the hit compounds from the organoid growth inhibition (OGI) screen, only EBET-1055 canceled the upregulation of all reporter activities in the co-culture, whereas GEM treatment did not. Because the reporter activity was compensated for by the viability of the LSCs, this effect was not simply due to growth inhibition or killing of the CAFs. Although BC2LCN+/Ki67+ cancer cells survived GEM treatment in vitro and in vivo (Supplementary Fig. 2), those cancer cells were very rare after EBET-1055 treatment of organoids (Fig. 1c). We therefore selected EBET-1055 as a seed compound, optimized it as a payload, and finally obtained the lead payload EBET-1593 (Supplementary Fig. 3b). For the conjugation to antibody, cathepsin-B-cleavable glycyl glycyl phenylalanyl glycine (GGFG) linker is linked to the hydroxyl group of EBET-1593 via self-immolative aminomethylene moiety.

### EBET-1055 works as a BET protein degrader, kills PDAC cells, and modulates CAF activity
EBET-1055 is a BET protein degrader that is composed of BET inhibitor (EBET-590), an E3 ubiquitin ligase ligand, and a linker (Supplementary Fig. 3b). This type of chimeric molecule, which forms a ternary

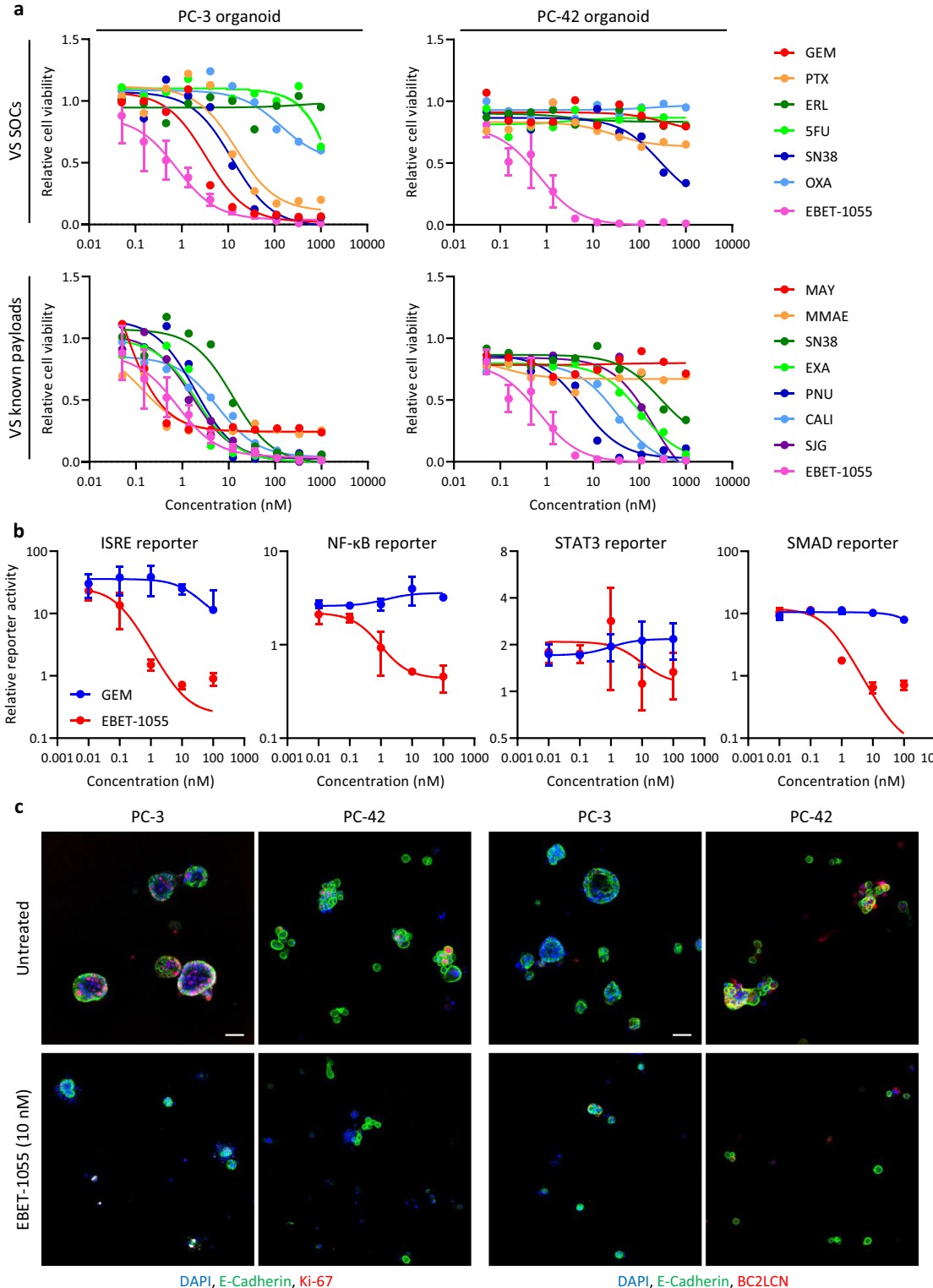

**Fig. 1 | Compound screening with organoids identifies EBET-1055, a compound effective against PDAC. a** PC-3 and PC-42 organoid growth inhibition by EBET and reference compounds. Data are presented as means ± standard deviation (*n* = 4 biological replicates for EBET, *n* = 2 biological replicates for others). Fitted curves with nonlinear regression are shown. GEM: gemcitabine; PTX: paclitaxel; ERL: erlotinib; 5FU: 5-fluorouracil; SN38: an active metabolite of irinotecan; OXA: oxaliplatin; MAY: maytansine; MMAE: monomethyl auristatin E; EXA: exatecan; PNU: PNU-159682; CALI: calicheamicin; SJG: SJG-136. **b** Inhibition of stromal signaling by

GEM and EBET in co-culture of PC-3 cells and liver stellate cells (LSCs). Signal intensity was normalized by the viability of LSCs, and relative values against the signal in monoculture of LSCs are presented as means ± standard deviation (*n* = 4 biological replicates). Fitted curves with nonlinear regression are shown. **c** Immunofluorescence staining of E-cadherin, BC2LCN, and Ki-67 in organoids. Representative images are shown from 2 independent experiments with similar results. Scale bars, 50 μm. **a, b** Source data are provided as a Source data file.

complex between target proteins and E3 ligases, induces potent degradation of target proteins via ubiquitination and subsequent destruction in proteasomes[8,9]. Live-cell monitoring of bromodomain-containing protein 4-bromodomain 1 (BRD4-BD1) domain degradation in HEK293 cells by using NanoBiT technology[23] showed that BET degradation by EBET-1055 occurred 10 min after addition and was saturated at 3 h (Supplementary Fig. 3c). The degradation efficacy of EBET derivatives and reference compounds at 3 h in this system was significantly correlated with their cell growth inhibition (CGI) efficacy against the AsPC-1 PDAC cell line (Supplementary Fig. 3d, e). The potency of EBET-1055 in OGI assay was much higher than dBET6, which is a BET protein degrader based on the well-known BET inhibitor JQ1[24] (Supplementary Fig. 3f).

Numerous studies have suggested that BET inhibitors possess anti-inflammatory or anti-fibrotic activity[25–28]; this suggestion is consistent with the inhibitory activity of EBET-1055 in the stromal signaling reporter system (Fig. 1b). To understand the underlying mechanism, we examined the status of the key signaling molecules STAT3 and SMAD2/3 by using mouse CAFs isolated from PDX tumors. STAT3 signaling mediates the phenotype of inflammatory CAFs (iCAFs), and SMAD signaling mediates the phenotype of myo-fibroblastic CAFs (myCAFs)[19–22]. Specific knockout of STAT3 on CAFs in a genetically engineered mouse PDAC model inhibits both the iCAF and myCAF phenotypes, affects immunosuppressive environment, induces anti-tumor immunity, and increases mouse survival, suggesting that STAT3 has a predominant role in the whole CAF population in PDAC[29]. EBET-1055 decreased the phosphorylation levels of STAT3$^{Y705}$, SMAD2$^{S465/467}$, and SMAD3$^{S423/425}$ 24 h after addition (Supplementary Fig. 4a). BRD proteins interact with STAT3 and SMAD3 and regulate their signaling[30,31], and BET inhibitors disrupt the interaction[31], inhibit myofibroblast differentiation[30], and decrease the phosphorylation levels of SMAD3 and STAT3 in mouse fibrotic kidney[26]. Our co-immunoprecipitation analysis also showed that BRD4 interacted with STAT3 on the transcriptional complex containing CDK9 in cells (Supplementary Fig. 4b), but the weak signal suggested that there was an indirect interaction through the multi-molecular complex. In the PDAC microenvironment, STAT3 signaling in iCAFs forms a positive feedback loop with the inflammatory cytokines IL-1, IL-6, leukemia inhibitory factor (LIF), and chemokine (C-X-C motif) ligand 1 (CXCL1) and orchestrates the immunosuppressive microenvironment[20]. In our analysis, co-culture of mouse CAFs with PC-3 or PC-42 cancer cells upregulated the secretion of mouse IL-6 and LIF from CAFs compared with monoculture of CAFs, and EBET-1055 treatment canceled the upregulation (Supplementary Fig. 4c).

## Establishment of CEACAM6 antibody, #84.7

For target selection of the ADC, we profiled gene expression in pancreatic cancer and normal tissues by using The Cancer Genome Atlas (TCGA[32]), Genotype-Tissue Expression (GTEx[33]), and Gene Expression Profiling Interactive Analysis (GEPIA[34]) databases. *CEACAM6* was the most upregulated gene encoding membrane protein in PDAC compared with in the normal pancreas. Furthermore, there were only eight membrane protein genes showing more than two-fold higher median transcripts per million (TPM) in PDAC than the highest median TPM in all normal tissues, and *CEACAM6* was one of them (Fig. 2a). Prognosis analysis revealed a correlation between *CEACAM6* expression and shorter survival (Fig. 2b). Immunohistochemical analysis revealed that CEACAM6 protein was expressed universally in clinical PDAC, and the expression was restricted to cancer cells (Fig. 2c and Supplementary Fig 5a). We then checked the relationships among *CEACAM6* expression, molecular subtypes of PDAC, and the expression of key molecules by using the TCGA PDAC dataset (Supplementary Fig. 5b). *GATA6* (GATA binding protein 6) is a

definitive endoderm specification gene, the expression of which is correlated with niche dependency, the classical subtype, and better prognosis of PDAC[35–37]. Platelet-derived growth factor receptor α/PDGFRα (*PDGFRA*) and α-smooth muscle actin/α-SMA (*ACTA2*) are markers of iCAF and myCAF, respectively[22]. *CEACAM6* expression was significantly greater in classical than in basal-like PDAC, and there was a weak but significant positive correlation between *CEACAM6* and *GATA6* expression (Supplementary Fig. 5c, d). In the selected PDAC-PDX models (a total of 16 models), all the CEACAM6$^{high}$ models were classical subtypes, and all the basal models were CEACAM6$^{low}$ (Supplementary Fig. 5e). CEACAM6 is expressed in normal lung epithelial cells and myeloid progenitor cells (MPCs)[38]; the results of our immunohistochemical analysis of lung tissue and our flow cytometric analysis of human MPCs differentiated from hematopoietic progenitor cells (HPCs) supported that expression (Fig. 2c and Supplementary Fig. 6a). We also conducted the absolute quantitation of CEACAM6 on cell membrane of cultured PDAC and normal cells (Supplementary Fig. 6b).

We developed antibodies against CEACAM6 from our proprietary phage display library of human single-chain variable antibody fragments (scFvs) and screened by flow cytometry using PDAC cells, human MPCs, and human small airway epithelial cells (HSAECs). We also examined cell-internalization activity by comparing mean florescence intensity of antibody-treated PC-42 cells after incubation under 4 and 37 °C culture conditions. Compared with the commercially available CEACAM6 monoclonal antibody (mAb) KOR, our clone #84.7 showed equivalent binding and stronger internalization activity to PC-42 cells, less binding to MPCs, and equivalent binding to HSAECs (Fig. 2d). Because #84.7 bound to CEACAM6 on human lung epithelial cells, we next chose to introduce an in vitro human lung epithelium culture called an ALI (air-liquid interface) culture[39]. We added ADCs into the lower chamber of this culture and examined the toxicity by ADCs of #84.7, epidermal growth factor receptor (EGFR) mAb (cetuximab), human epidermal growth factor receptor 2 (HER2) mAb (trastuzumab), and PNU-159682 (the most potent reference payload in our OGI screening; Fig. 1a). Antibody against hen egg lysozyme 3 (HEL3) was used as a non-targeting control antibody. #84.7 had comparatively low toxicity to lung epithelium (Fig. 2e). Immunohistochemical analysis of the ALI culture and immunofluorescence analysis of monkey lung showed that CEACAM6 expression was restricted to the apical side of the lung epithelium, whereas EGFR and HER2 expression was distributed over the whole epithelium (Supplementary Fig. 6c). Additionally, we examined the in vivo distribution of injected antibodies in cynomolgus monkeys. There was no great difference in the half maximum effective concentration of #84.7 binding to human and cynomolgus CEACAM6 expressed on HEK293 cells (Supplementary Fig. 6d). By visualization of injected mAbs by staining with anti-human IgG antibody, we found that #84.7 had only a background level of distribution on the bronchial epithelium despite the CEACAM6 expression of this epithelium, whereas EGFR mAb (cetuximab) and HER2 mAb (trastuzumab) accumulated on the bronchial epithelium with target expression (Fig. 2f). There was no distribution of these antibodies on the alveoli. These results suggest that #84.7 was not actively distributed to CEACAM6 on the apical side of the lung epithelium, presumably because of the barrier function of the epithelium tight junctions.

## 84-EBET efficiently kills various PDAC organoids and modulates CAF activity via the bystander effect

We examined the in vitro efficacy of EBET-1593-conjugated #84.7 (84-EBET, drug-antibody ratio 4) in a PDAC-OGI assay derived from 16 PDAC-PDX models. We also evaluated #84.7 conjugated with MMAE or deruxtecan (DXd), both of which are clinically approved payloads in solid cancers. 84-EBET had a lethal effect in most of the models,

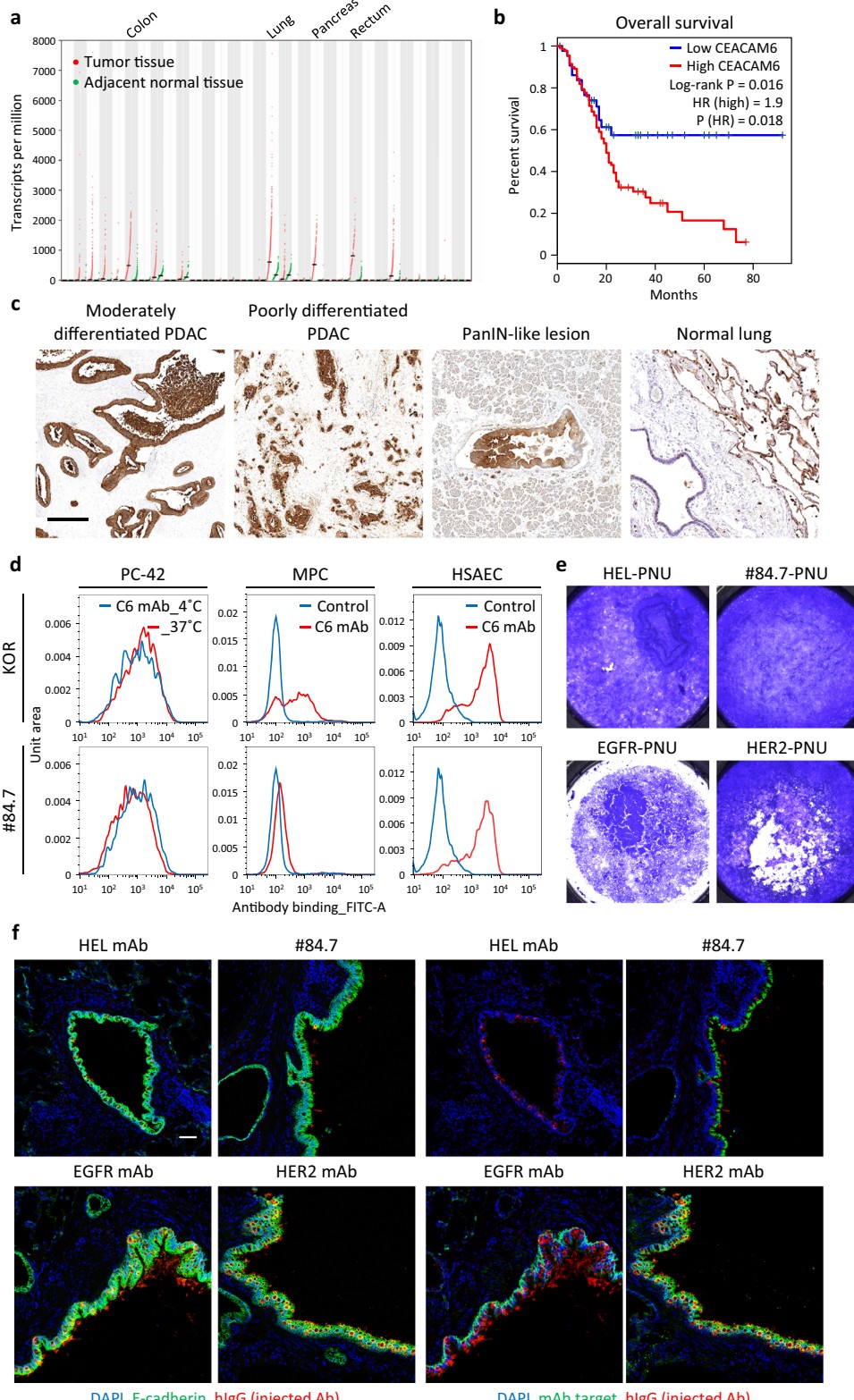

DAPI, E-cadherin, hIgG (injected Ab)        DAPI, mAb target, hIgG (injected Ab)

whereas 84-MMAE and 84-DXd did not in all the models (Fig. 3a). Most of the CEACAM6$^{low}$/basal models were relatively resistant to 84-EBET treatment, but one CEACAM6$^{low}$/basal model, KYK-048, showed supersensitivity to 84-EBET treatment. We examined the in vitro bystander efficacy of 84-EBET in a co-culture of CEACAM6-expressing PDAC cells (target, HPAF-II/Cas9/Fluc (firefly luciferase)) and CEACAM6-knockout PDAC cells (bystander, HPAF-II/CEACAM6-KO/ Rluc (renilla luciferase)). The viability of the bystander cells (activity of Rluc) decreased in the presence of the target cells after treatment with CEACAM6-ADC with bystander efficacy. 84-EBET showed stronger bystander efficacy than 84-MMAE and 84-DXd (Fig. 3b). We then examined whether 84-EBET modulated CAF activity via bystander efficacy. In a co-culture assay of PDAC cells and mouse CAFs, 84-EBET decreased mouse IL-6 and LIF secretion by CAFs but did not affect the viability of CAFs at these concentrations (Fig. 3c, d and Supplementary Fig. 6e).

**Fig. 2 | Establishment of CEACAM6 antibody, #84.7. a** Expression of *CEACAM6* mRNA in 33 cancer types (red dots) and paired normal tissues (green dots) was plotted by using the TCGA database and GEPIA server. Bars represent means. Arrowhead indicates PDAC data. **b** Patient survival analysis in PDAC patients (*n* = 90, from TCGA) according to *CEACAM6* mRNA expression (high: upper 75%; low: lower 25%). HR, hazard ratio; 95% confidence interval. *P*-value calculated by log-rank test is shown on the plot. **c** Immunohistochemical staining of CEACAM6 in human PDAC samples and normal lung tissue. Representative images are shown from 2 independent experiments with similar results. Scale bar, 100 μm. PanIN, pancreatic intraepithelial neoplasia. **d** Flow cytometric analysis of CEACAM6 antibodies using PC-42 cancer cells, myeloid progenitor cells (MPCs) and human small airway epithelial cells (HSAECs). With PC-42 cells, cell-internalization activity was

also examined after incubation in culture at 37 ˚C. Representative data are shown from 2 independent experiments with similar results. KOR is a reference mAb for CEACAM6. **e** In vitro lung toxicity assay with air-liquid interface (ALI) culture of human pulmonary alveolar epithelial cells. Five days after incubation of ADCs in the lower chamber of ALI culture in 1 nM of PNU-conjugated #84.7, hen egg lysozyme 3 (HEL) mAb, EGFR mAb (cetuximab), or HER2 mAb (trastuzumab), surviving epithelial cells were stained with crystal violet. Representative data are shown from 2 independent experiments with similar results. **f** Immunofluorescence staining of injected antibodies in monkey lung. Twenty-four hours after injection of #84.7, HEL mAb, EGFR mAb (cetuximab), or HER2 mAb (trastuzumab), injected antibodies were visualized by using anti-human-IgG, and the respective antibody targets were visualized by using antibodies different from the injected ones. Scale bar, 50 μm.

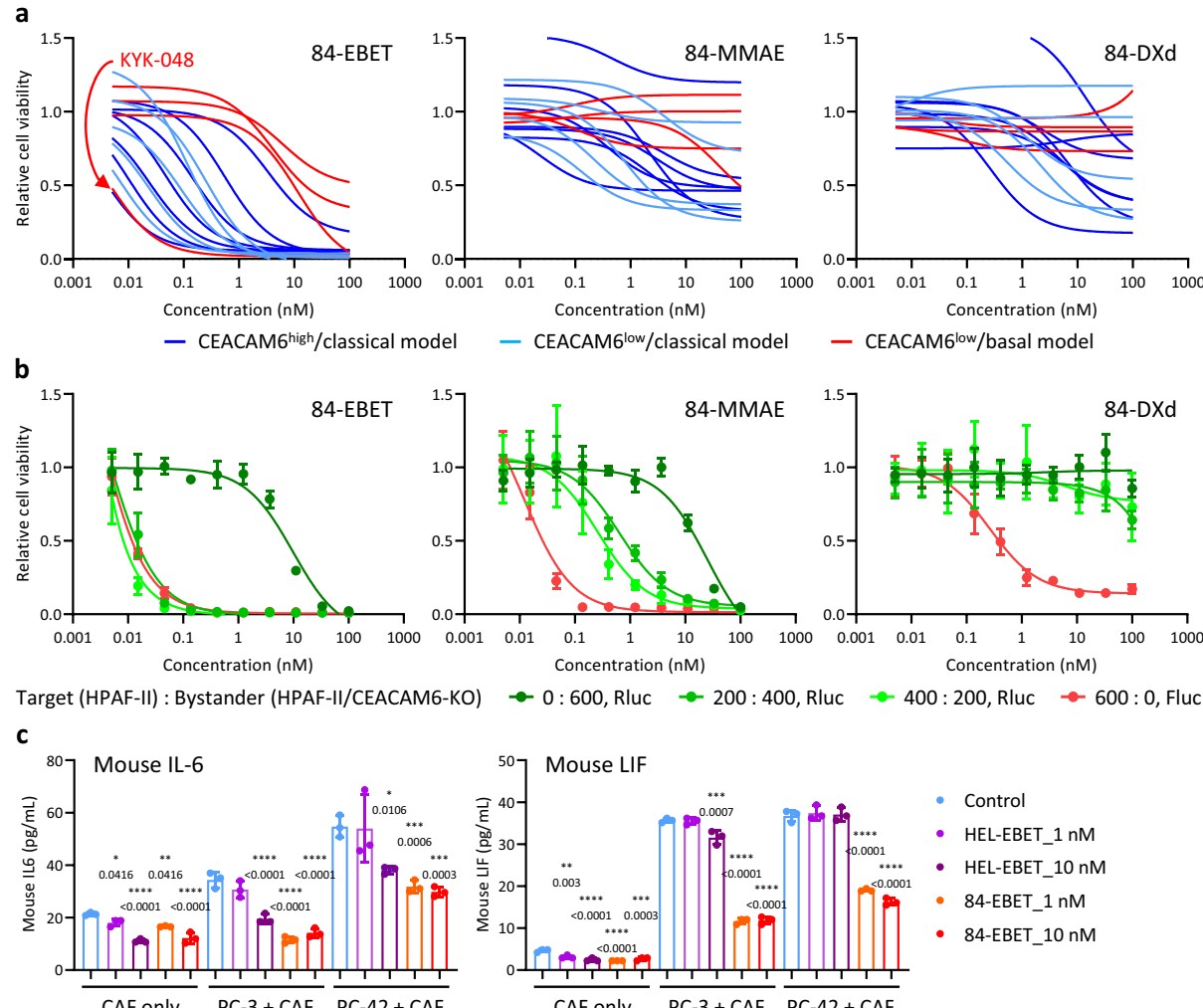

**Fig. 3 | 84-EBET efficiently kills various PDAC organoids and modulates CAF activity via the bystander effect. a** Organoid growth inhibition by 84-EBET, 84-MMAE, and 84-DXd (-deruxtecan), in 16 PDAC organoid models (*n* = 4 biological replicates). Fitted curves with nonlinear regression are shown. **b** Cell growth inhibition by 84-EBET, 84-MMAE, and 84-DXd in co-culture of HPAF-II/Cas9/Fluc (target) and HPAF-II/CEACAM6-KO/Rluc (bystander) cells at multiple ratios. The viabilities of the target and bystander were measured by using dual luciferase assay.

Data are presented as means ± standard deviation (*n* = 4 biological replicates). Fitted curves with nonlinear regression are shown. **c** Quantification of inflammatory cytokines released from CAFs in co-culture with PC-3 and PC-42 cells. Data are presented as means ± standard deviation (*n* = 3 biological replicates). **P* < 0.05, ***P* < 0.01, ****P* < 0.001, *****P* < 0.0001, one-way ANOVA test followed by Dunnett's test between control group and treatment groups in each culture. **a**–**c** Source data are provided as a Source data file.

## 84-EBET induces substantial regression of various PDAC-PDX tumors after a single injection

We examined the in vivo efficacy of 84-EBET in 16 PDAC-PDX models. In some models, 84-MMAE, 84-DXd, or trophoblast cell-surface antigen 2 (Trop2) antibody conjugated with EBET-1593 (Trop2-EBET) was

also evaluated. A 3 mg/kg dose of 84-EBET was half of the maximum tolerated dose in mice. A single injection of 3 mg/kg of 84-EBET induced substantial tumor regression in all the classical models, without significant body weight loss (Fig. 4a, b and Supplementary Fig. 7). In 9/16 PDX models, tumor regression was maintained for at

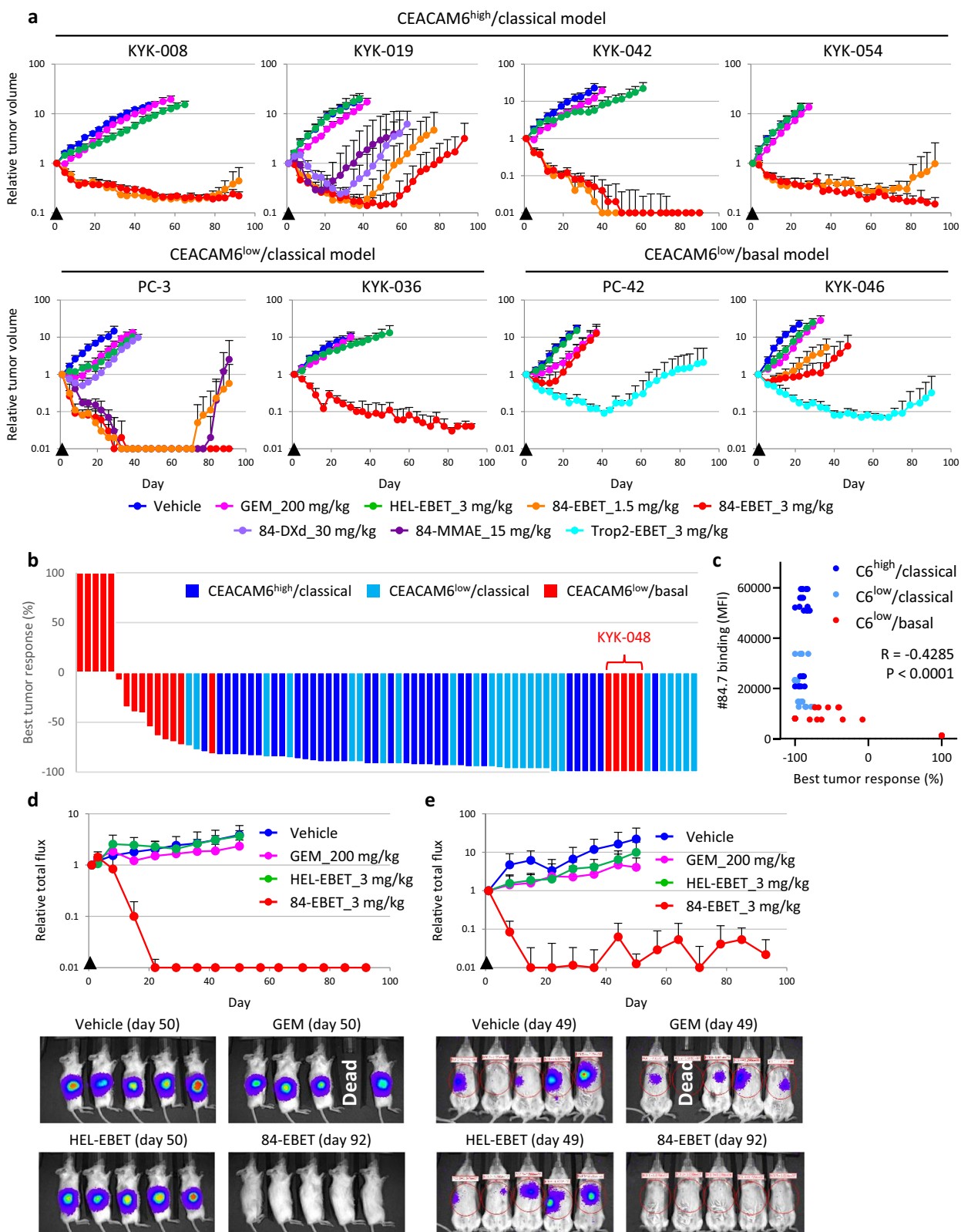

least 3 months. 84-DXd and 84-MMAE showed weaker efficacy than 84-EBET in the KYK-019 and PC-3 models, although the doses were higher than those reported in preclinical mouse studies of HER2-DXd[40–44] and Nectin-4-MMAE[45]. Notably, in the CEACAM6low/basal PC-42 and KYK-046 models, the efficacy of 84-EBET was limited, but the same dose of Trop2-EBET induced more profound regression, indicating that EBET was effective provided that it was delivered with the appropriate targeted antibody. In another CEACAM6low/basal model, KYK-048, 84-EBET induced 100% tumor shrinkage; this was consistent with the supersensitivity to 84-EBET treatment in the OGI assay (Fig. 3a). The degree of binding of #84.7 to freshly isolated PDAC cells was significantly correlated with the best tumor response (Fig. 4c). Subsequently, we examined the in vivo efficacy of 84-EBET by using orthotopic and liver metastasis models. In these models, PDX-derived

**Fig. 4 | 84-EBET induces substantial regression of various PDAC-PDX tumors after a single injection. a** Representative tumor growth curves in response to GEM and 84-EBET in the PDAC-PDX panel. 84-MMAE, 84-DXd, and Trop2-EBET were examined in only some models. The average starting tumor volumes were as follows, PC-3: 100 mm³, PC-42: 80 mm³, KYK-008: 100 mm³, KYK019: 100 mm³, KYK-036: 140 mm³, KYK-042: 75 mm³, KYK-046: 80 mm³, KYK-054: 100 mm³. Arrowheads indicate times of drug administration. Complete regression (100% reduction of tumor volume) was defined as 0.01 on a log-scale graph. Data are presented as means ± standard deviation (*n* = 5 mice). **b** Waterfall plot showing the best tumor response after a single injection of 84-EBET at 3 mg/kg. The average starting tumor volumes ranged between 75 and 140 mm³. Tumor volume changes from the baseline of 16 PDAC-PDX models are shown (*n* = 5 mice). **c** Binding affinity of #84.7 to PDAC cells in each model was quantitated by flow cytometry and plotted against

tumor response (*n* = 5 mice). R-value and *P*-value calculated by using Pearson's correlation (two-sided) are shown on the plot. MFI, mean fluorescence intensity. **d** Tumor growth curves in response to GEM and ADCs with 84-EBET in the orthotopic transplant of PC-3 model. Arrowheads indicate times of drug administration. Complete regression (100% reduction of Fluc signal) was defined as 0.01 in a log scale graph. Data are presented as means ± standard deviation (*n* = 5 mice). Representative luminescence images are shown in the bottom panels. **e** Tumor growth curves in response to GEM and ADCs with 84-EBET in liver metastasis in the PC-3 model. Arrowheads indicate times of drug administration. Complete regression (100% reduction of Fluc signal) was defined as 0.01 on a log-scale graph. Data are presented as means ± standard deviation (*n* = 6 mice). Representative luminescence images are shown in the bottom panels. **a**–**e** Source data are provided as a Source data file.

PC-3 cancer cells were infected with Fluc-expressing lentivirus and injected into mouse pancreas or portal vein. After a single injection of 84-EBET, all the established tumors became nearly undetectable for at least 3 months (Fig. 4d, e). As opposed to the subcutaneous PC-3 model (Fig. 4a), these models showed only growth delay upon GEM treatment.

### 84-EBET modulates CAFs via the bystander effect

We examined markers of 84-EBET pharmacodynamics by using the PC-3 and PC-42 models. Because there are many reports on CAF markers, we used single cell RNA-seq data to determine which markers would be appropriate in our models. After single-cell RNA-seq analysis of non-treated PC-3 and PC-42 tumors, the data from mouse stromal cells were mixed and clustered in a t-distributed stochastic neighbor embedding (t-SNE) plot (Supplementary Fig. 8a). We then plotted the expression of reported CAF markers, namely, for iCAF, *PDGFRA*/PDGFRα; for myCAF, *ACTA2*/α-SMA; and for panCAF, *COL1A1*/collagen I. The results suggested that cluster A was an iCAF-like population and cluster B was a myCAF-like population. Consistently, gene set enrichment analysis (GSEA) showed that hallmark gene sets of IL6-JAK-STAT3 signaling, TNFα signaling via NF-κB, and the inflammatory response were enriched in cluster A compared with cluster B (Supplementary Fig. 8b, c). These data indicate that these markers can be appropriate pharmacodynamics markers of CAF subpopulations. Immunohistochemical analysis of PDX tumors confirmed that BRD4 was degraded by 84-EBET on both cancer and stromal cells in the tumors, suggesting that EBET diffused from the cancer cells had a bystander effect on CEACAM6-negative stromal cells (Fig. 5a, b). 84-EBET treatment also decreased the pSTAT3^Y705- and PDGFRα-positive areas in the stroma with statistical significance (Fig. 5a–c). The effects on pSMAD2/3 and α-SMA, which are markers of myCAFs, lacked consistency at these time points. We also profiled CAF-related gene expression in mouse stromal cells by using bulk RNA-seq data on PC-3 and PC-42 tumors. Treatment with 84-EBET reduced the expression of most of the iCAF-related genes in the stroma in both models, whereas GEM treatment down-regulated only a few (Supplementary Fig. 9a). Especially, the down-regulation of a wide range of inflammatory cytokines supported the results of the in vitro co-culture assay (Fig. 3c). Additionally, GSEA by using the mouse transcriptome data supported the suppression of iCAF-related stromal signaling (Supplementary Fig. 9b).

To clarify the causal relationship of the effects of 84-EBET on CAFs and the immune system, we established an in vitro tri-culture system of PC-3 cancer cells transduced with Fluc, mouse CAFs, and human peripheral blood mononuclear cells (PBMCs). In this system we observed the killing of PC-3 cancer cells by PBMCs, but it was suppressed by co-culture with CAFs (Supplementary Fig. 10a). Treatment with anti-human-PD-1 mAb, an immune checkpoint inhibitor (ICI), dose-dependently enhanced the killing effect in the absence of CAFs but did not in the presence of CAFs. Quantitation of IL-2 and IFN-γ in the culture supernatant showed that CAFs significantly inhibited T-cell

activation among PBMCs (Supplementary Fig. 10b). Finally, we examined ADC treatment in this system and found that 84-EBET treatment canceled the suppression of T-cell-mediated killing by CAFs (Supplementary Fig. 10c). These data suggested that EBET was released from CEACAM6-expressing PDAC cells, diffused to neighboring CAFs, and canceled immune suppression by the CAFs.

### 84-EBET has a combinational effect with standard chemotherapy and PD-1 antibody, without additional body weight loss

We examined the strategy of combining 84-EBET with standard chemotherapy in a CEACAM6^low/basal PC-42 model. Combination of 84-EBET (3 mg/kg) with standard chemotherapy for PDAC, namely GEM + nab-PTX, induced more profound and sustained tumor regression, without toxicity enhancement as manifested by a lack of bodyweight change (Fig. 6a). Next, to evaluate the combination with an ICI, we established a human CEACAM6 (hCEACAM6)-expressing mouse syngeneic PDAC model[46], Pan02/hCEACAM6. PD-1 antibody alone induced only growth delay, and 84-EBET at 1.5 mg/kg led to tumor relapse in this model (Fig. 6b). However, a combination of 84-EBET and PD-1 antibody maintained complete regression of all tumors for at least 3 months. An immunohistochemical analysis of Pan02 tumors revealed that 84-EBET treatment had reduced the area of PDGFRα-positive stroma in the tumor by day 3 and subsequently increased activated T-cell infiltration by day 7 (Fig. 6c and Supplementary Fig. 11). Consistent with the in vitro data (Supplementary Fig. 10c), this suggested that 84-EBET inhibited the inflammatory phenotype of CAFs and reversed the immunosuppressive tumor microenvironment. Immune profiling by mass cytometry on day 7 revealed that 84-EBET decreased immunosuppressive cells in tumor, myeloid-derived suppressor cells (MDSCs) and tumor-associated macrophages (TAMs), and increased effector cells in tumor, T cells and natural killer (NK) cells (Supplementary Fig. 12). In tumor-draining lymph nodes, mature dendritic cells (DCs) and primed T cells increased in the combination group only, suggesting enhanced antigen presentation in tumor. Taken together, our hypothesis is that there are two phases of effects on the immune system. In the first phase, CAF modification by 84-EBET treatment inhibits inflammatory response and immunosuppressive cell infiltration, attracts and activates effector cells, and exerts an antitumor effect. In the next phase, cancer cells die, antigen uptake by dendritic cells is enhanced, and activated dendritic cells migrate to their lymph nodes to promote further T cell activation.

## Discussion

The pharmacological profile of our CEACAM6-EBET ADC can be summarized in three points. The first is BRD degradation in CEACAM6-positive PDAC cells by the delivery of EBET using CEACAM6 antibody. BRD degradation was able to kill almost all PDAC cells. The second point is the modulation of PDAC stromal signaling via bystander efficacy on CEACAM6-negative CAFs. STAT3, a key signaling molecule in CAFs, is functionally coupled to BRD2/4, and diffused EBET payloads from PDAC

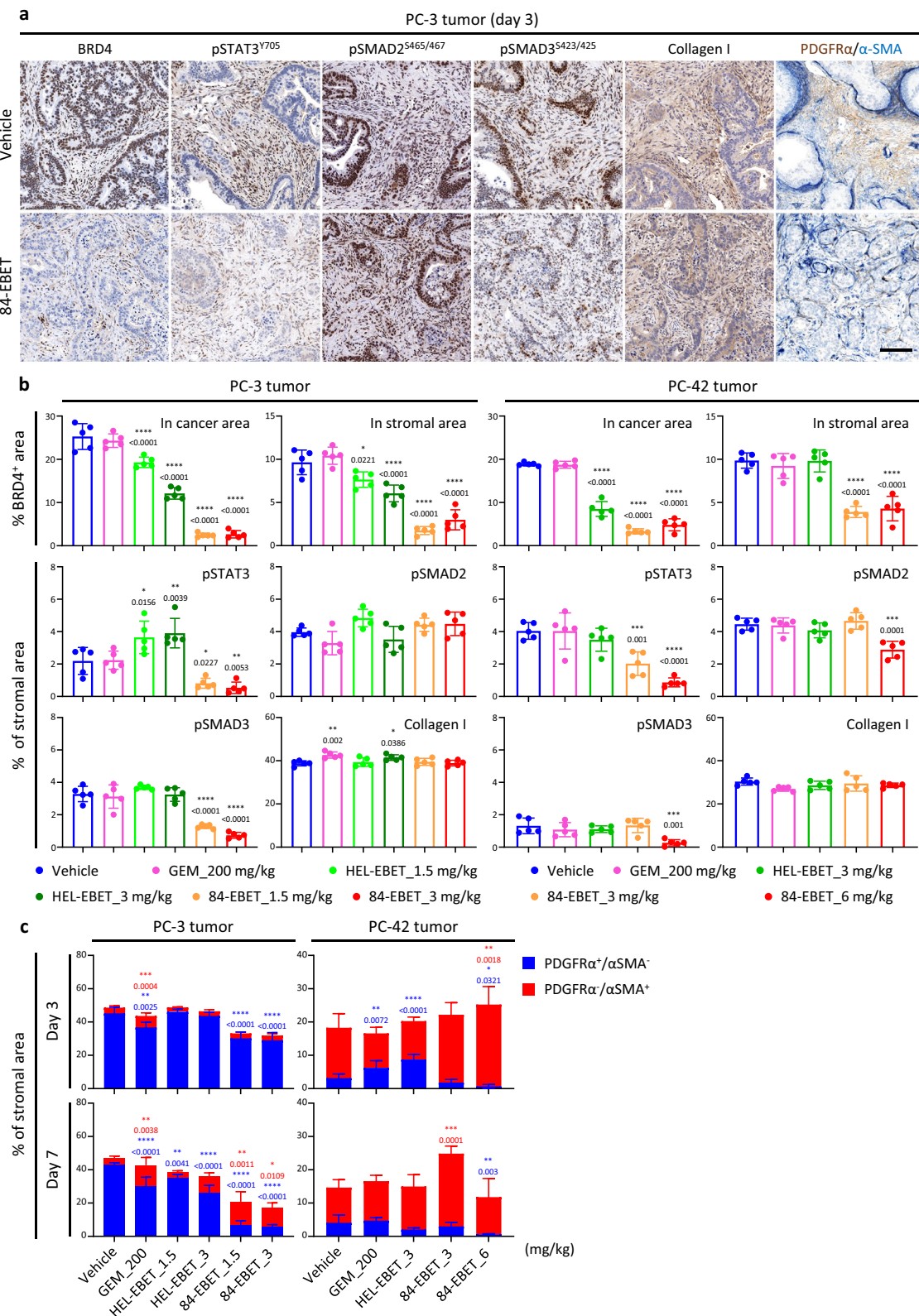

cells inhibit STAT3 signaling in CAFs. Although EBET might kill CAFs eventually, our analyses suggested that functional inhibition of CAFs and modulation of PDAC stromal signaling in vivo are consequences of CEACAM6-EBET ADC treatment. We believe that the combination of efficient killing of PDAC cells and modulation of the PDAC stroma is the central and most important feature of CEACAM6-EBET ADC. Many clinical studies have been exploring the potential of STAT3 and TGF-β

signal inhibitors targeting the tumor stroma or microenvironment, but so far the efficacy of these inhibitors has been very limited[47,48]. The third point is the biological profile of our CEACAM6 antibody #84.7. One study has reported the distribution and toxicity of ADC with CEACAM6 mAb and maytansinoid (CEACAM6-mAb-DM1) in monkeys[38]. The authors found that their CEACAM6 mAb recognized MPCs, intestinal mucosa, and lung alveoli, but it was distributed specifically to the bone

**Fig. 5 | 84-EBET modulates CAFs via the bystander effect.**
**a** Immunohistochemical staining for BRD4, pSTAT3, pSMAD2, pSMAD3, collagen I, PDGFRα, and α-SMA in the PC-3 tumor 3 days after treatment with 84-EBET. Representative images are shown from 2 independent experiments with similar results. Scale bar, 100 μm. **b** Quantitation of BRD4, pSTAT3, pSMAD2, pSMAD3, and collagen I staining in the PC-3 and PC-42 tumors 3 days after drug treatment. The cancer cell area and stromal cell area were distinguished by nuclear shape using Indica Labs' HALO system. The ratio of BRD4$^+$ area to total area was calculated in cancer cell nuclei and in stromal cell nuclei. The ratio of pSTAT3$^+$, pSMAD2$^+$, pSMAD3$^+$, and collagen I$^+$ stromal area to total stromal area was calculated. Data are presented as means ± standard deviation ($n = 5$ mice). *$P < 0.05$, **$P < 0.01$, ***$P < 0.001$, ****$P < 0.0001$, one-way ANOVA test followed by Dunnett's test between vehicle-treated group and drug-treated groups. **c** Quantitation of PDGFRα and α-SMA staining in the PC-3 and PC-42 tumors 3 and 7 days after drug treatment ($n = 5$). The ratio of PDGFRα$^+$ and α-SMA$^+$ stromal area to total stromal area was quantified. Data are presented as means ± standard deviation ($n = 5$ mice). *$P < 0.05$, **$P < 0.01$, ***$P < 0.001$, ****$P < 0.0001$, one-way ANOVA test followed by Dunnett's test between vehicle-treated group and drug-treated groups. **b**, **c** Source data are provided as a Source data file.

marrow after intravenous injection. Consistently, CEACAM6-mAb-DM1 induced only neutropenia. #84.7 showed less binding than the reference CEACAM6 mAb to MPCs and carried a lower lung toxicity risk than EGFR and HER2 mAbs (Fig. 2d–f). CEACAM6 is a highly glycosylated protein, and there have been reports of mAbs recognizing the CEACAM6 glycan structure[49,50]. #84.7 might recognize a different glycan structure on PDAC cells from that on blood cells. Our finding that CEACAM6 expression was restricted to the apical side of the lung epithelium is a possible reason for the lower lung toxicity (Supplementary Fig. 6c). Furthermore, after evaluation of the lead ADC with EBET-1593, final optimization of payload has been conducted especially focusing assays with human lung epithelium, HPCs, and MPCs to improve the tolerability. The final ADC candidate with EBET-2113 did not show any effect on their viability at 3.3 nM but showed lethal effect on PDAC cells at 0.1 nM (Supplementary Fig. 13a–c). The ADC is currently in process development for GLP tolerability testing with monkeys.

Cancer immunotherapy (called immuno-oncology (IO) therapy) boosts the intrinsic immune system to fight cancer. Its tools include cancer vaccines, adoptive cell transfer, oncolytic viruses, and ICIs. Although IO therapies, especially ICIs, have greatly improved the survival of some patients with certain cancer types, a substantial percentage of patients fail to respond or progress. To tackle this problem, a great number of combinational ICI treatments are being tested in clinical trials[51]. Among the various cancers resistant to ICIs, those lacking T cells because of the presence of physical and chemical barriers mediated by stromal cells are thought to be the hardest ones to treat. The gene signature of activated stroma and extracellular matrix accumulation is associated with poor outcomes of PD-1 blockade[52,53]. PDAC has shown the highest score for activated stroma and has had disappointing results from early clinical trials of ICIs[54]. Our organoid system was developed to evaluate the complexity of cancer and stroma in PDAC and would be a powerful tool for seeking the next breakthrough. With this system, we obtained a compound, EBET-1055, that was effective in both PDAC cells and the microenvironment.

Accordingly, here we show the potential efficacy of pancreatic cancer by CEACAM6-EBET ADC powered by its cancer-cell-killing activity, stromal modulation activity, and good antibody profile.

## Methods
All the animal experiments were conducted in accordance with the Institutional Animal Care and Use Committee guidelines of Eisai Co., Ltd. and Shin Nihon Kagaku, Ltd. (animal study protocols of Eisai v18, SBL038-158). All the experiments with human samples were approved by the ethics committee of Eisai Co., Ltd. and Osaka National Hospital (REP-2019-0552, REP-2017-0445, REP-2017-0166, #2007-0071).

### Development of CEACAM6 antibody
scFv clones with affinity for CEACAM6 were selected from the phage display library of human scFv (Kan Libs, KAN Research Institute) by using a recombinant protein of the extracellular domain of CEACAM6 (a.a. 33–314) purified from the Expi293F system (Thermo Fisher). During the selection, clones with affinity for CEACAM1 and CEACAM5 were discarded. The selected scFv clones were then converted to human IgG and screened by flow cytometry with various cells expressing CEACAM6.

### Development of EBET compound
ER-001251206-000 (=EBET-590), ER-001326054-000 (=EBET-1055, seed payload), ER-001388472-000 (= EBET-1593, lead payload), and ER-001426363-000 (=EBET-2113, final payload) were synthesized at Eisai Co., Ltd. in accordance with procedures in our patent (application number: 63/373646, 63/377518). The screening flow of payloads was as follows in the following order: CGI assays with HPAF-II and AsPC-1 PDAC cell lines, analysis of physicochemical properties, ADC conjugation with various linkers, CGI assays with HPAF-II, HPAF-II/CEACAM6-KO, AsPC-1 and AsPC-1/CEACAM6-KO cells, OGI assays with PC-3 and PC-42 PDAC-PDX models, in vitro bystander efficacy assay, in vitro plasma stability assay, and in vivo tumor growth inhibition assay with HPAF-II, AsPC-1, PC-3 and PC-42 models. In the process of final optimization after EBET-1593, in vitro killing assays with human lung epithelium, HPCs and MPCs and rat tolerability assay were added.

**Preparation of compound.** GEM, PTX, erlotinib, 5FU, SN38, oxaliplatin, maytansine, MMAE, exatecan, PNU-159682, chalicheamicin, SJG-136, JQ1, and dBET6 were purchased from Astatech, FUJIFILM Wako Pure Chemical Corporation, LKT Laboratories, Sigma-Aldrich, Shanghai Sunway Pharmaceutical Technology, Xdcexplorer, Concortis Biosystems, Shanghai Haoyuan Chemexpress, Amadis Chemical, and MedChemexpress. JQ1 was synthesized at Piramal Enterprises by using a procedure in a published patent (CN103694253).

### Preparation of antibody–drug conjugates
Cathepsin-B-cleavable glycyl glycyl phenylalanyl glycine (GGFG) linker is linked to the hydroxyl group of EBET via self-immolative aminomethylene moiety in accordance with procedures in our patent (application number: 63/373646, 63/377518). PNU-159682, DXd, and MMAE linker payloads were synthesized at Eisai Co., Ltd. in accordance with procedures in published patents (WO2009099741, WO2015095223, WO2019044947, WO2003043583). Cetuximab (EGFR mAb) and trastuzumab (HER2 mAb) were purchased from Merck and Chugai. Control IgG (HEL3 mAb) and sacituzumab (Trop2 mAb) were provided by KAN Research Institute and Epochal Precision Anti-Cancer Therapeutics (EPAT) respectively. All ADCs are prepared by the following general procedure for cysteine-based conjugation. Antibodies were reduced by Tris(2-carboxyethyl)phosphine at room temperature overnight. The reduced antibodies were diluted with 50% propylene glycol in 1 mM EDTA-Dulbecco's phosphate-buffered saline, and then maleimide-containing linker payloads were added. After incubation for 90 min at room temperature, the reaction mixture was concentrated and buffer-exchanged by using an Amicon Ultra-15 (30 K) centrifugal filter unit, with sterile filtration. The ADC was analyzed by using size-exclusion chromatography with Xbridge Protein BEH SEC 200-Å columns (Waters), hydrophobic interaction chromatography with TSKgel Butyl-NPR columns (Tosoh), and liquid chromatography–mass spectrometry to determine the drug to antibody ratios.

### Acquisition of PDAC-PDX models
KYK models (KYK-008, 017, 019, 031, 036, 042, 046, 048, 054, 060, 062, 065, 069, 090) were obtained from KAN Research Institute and PC models (PC-3, 42) were obtained from The Tsukuba Human Tissue

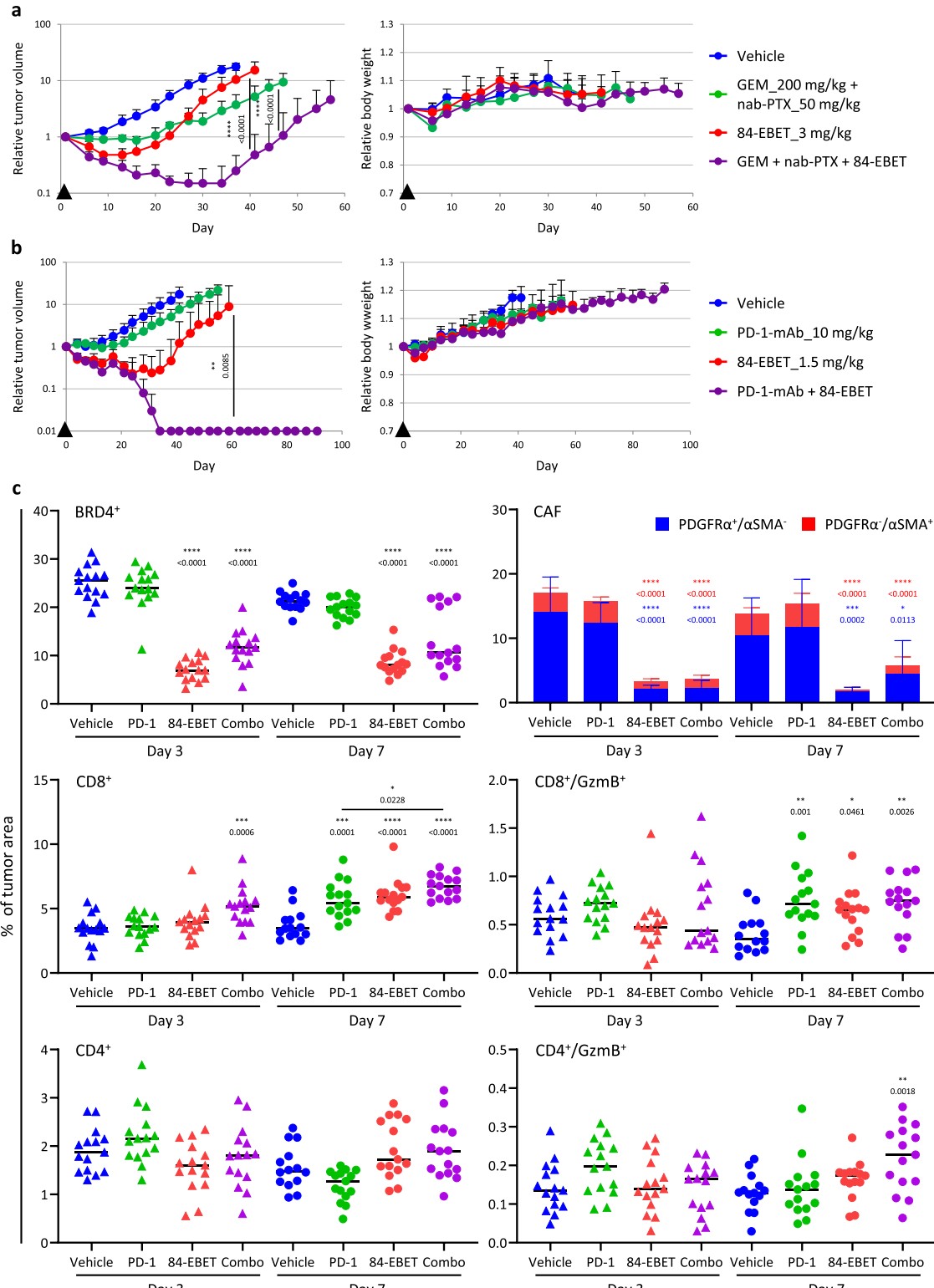

**Fig. 6 | 84-EBET has a combinational effect with standard chemotherapy and PD−1 antibody, without toxicity enhancement. a** Tumor growth curves and body weight curves in response to GEM + nanoparticle albumin-bound paclitaxel (nab-PTX), 84-EBET, and the combination of both in the PC-42 model. The average starting tumor volume was 80 mm³. Arrowheads indicate times of drug administration. Data are presented as means ± standard deviation (*n* = 5 mice). ****P < 0.0001, two-way ANOVA test followed by Fisher's LSD test. **b** Tumor growth curves and body weight curves in response to PD-1-mAb, 84-EBET, and the combination of both in the Pan02/hCEACAM6 model. The average starting tumor volume was 80 mm³. Complete regression (100% reduction of tumor volume) was defined as 0.01 on a log-scale graph. Arrowheads indicate times of drug administration. Data are presented as means ± standard deviation (*n* = 5 mice). **P < 0.01, two-way ANOVA test followed by Fisher's LSD test. **c** Quantitation of BRD4, PDGFRα, α-SMA, CD8, and granzyme B (GzmB) staining in the Pan02/hCEACAM6 tumors 3 and 7 days after drug treatment. The ratios of BRD4⁺, PDGFRα⁺, α-SMA⁺, CD8⁺, CD4⁺, and GzmB⁺ area to total tumor area were calculated by using Indigo Labs' HALO system. Data are presented as means ± standard deviation (*n* = 15 mice). *P < 0.05, **P < 0.01, ***P < 0.001, ****P < 0.0001, one-way ANOVA test followed by Dunnett's test between vehicle-treated group and drug-treated groups. Combo, combination of 84-EBET and PD-1-mAb. **a**–**c** Source data are provided as a Source data file.

Biobank Center. All the models are made available through a material transfer agreement with KAN Research Institute or The Tsukuba Human Tissue Biobank Center.

## Preparation of PDAC cell lines, PDX-derived PDAC cells, and primary normal cells

HPAF-II (CRL-1997) and AsPC-1 (CRL-1682) cell lines were purchased from ATCC and cultured in RPMI1640 medium containing 10% fetal bovine serum (FBS). The Pan02 (CVCL_D627) cell line was purchased from the National Institutes of Health and cultured in Eagle's minimal essential medium with 10% FBS. Human LSCs (#5300) and PSCs (#3830) were purchased from ScienCell Research Laboratories and cultured in Stellate Cell Medium (ScienCell Research Laboratories). Mouse CAFs were isolated from PC-3 subcutaneous tumors by using a Mouse Tumor-Associated Fibroblast Isolation Kit (Miltenyi Biotech) and human CD326/EpCAM MicroBeads (Miltenyi Biotech) in accordance with the manufacturer's instructions. Isolated CAFs were cultured in MSCGM (mesenchymal stem cell growth medium) (Lonza). Human PBMCs (CC-2704) were purchased from Lonza and cultured in ImmunoCult-XF T Cell Expansion Medium (STEMCELL Technologies) supplemented with 12.5 ng/mL human IL-7 (Miltenyi Biotec), 12.5 ng/mL human IL-15 (Miltenyi Biotec), and T Cell TransAct (Miltenyi Biotec). Human umbilical cord blood CD34$^+$ HPCs (2C-101) were purchased from Lonza and cultured in StemSpan SFEMII with StemSpan CD34$^+$ Expansion Supplement (STEMCELL Technologies). PC-3 and PC-42 cancer cells were isolated from PC-3 and PC-42 subcutaneous tumors. Briefly, tumors were extirpated from mice and digested with a Tumor Dissociation Kit and gentleMACS Dissociator (Miltenyi Biotec). Isolated single cells were cultured in DMEM/F12 supplemented with 10% embryonic stem-cell FBS (Thermo) after depletion of mouse stromal cells by using a Mouse Cell Depletion Kit (Miltenyi Biotech). For PC-3 culture, the following inhibitor cocktail (4i cocktail) was added to the culture: 1 μM A83-01 (Wako), 10 μM Y-27632 (MedChemExpress), 1 μM CHIR-99021 (Cayman Chemical), and 1 μM DMH-1 (Shanghai Haoyuan Chemexpress). Normal HSAECs (CC-2547) were purchased from Lonza, and normal human pulmonary alveolar epithelial cells (HPAEpiCs, #3200) were purchased from ScienCell Research Laboratories. These lung epithelial cells were cultured in PneumaCult-Ex Plus media (STEMCELL Technologies) supplemented with 4i cocktail. For cell authentication, primary human PDAC cells were examined by sequencing oncogenes, and mouse CAFs were examined by fibroblast marker expression. All cell lines were tested negative for mycoplasma contamination.

## Lentiviral infection and cell establishment

The lentiviral vector expressing Fluc linked via 2A peptide with enhanced green fluorescent protein was constructed by using pSMPUW-Puro lentivirus expression vector (Cell Biolabs). The lentiviral vector expressing Rluc linked via 2A peptide with turbo red fluorescent protein was constructed by using pCDH-MSCV-MCS-EF1α-Puro lentivirus expression vector (System Biosciences). The lentiviral vector expressing human or cynomolgus CEACAM6 was constructed by using pLV[Exp]-Puro-EFS lentiviral vector (VectorBuilder). The lentiviral vector expressing LgBiT[23] was constructed by using pLV[Exp]-Bsd-EFS lentiviral vector (VectorBuilder). The lentiviral vector expressing the human BRD4-BD1 domain (a.a. 1–209) tagged with HiBiT[23] peptides at the N-terminus (BD1-HiBiT) was constructed by using pLV[Exp]-Puro-hPGK lentiviral vector (VectorBuilder). VSV-G pseudotyped lentivirus particles were produced by using MISSION Lentiviral Packaging Mix (Sigma-Aldrich) and used to infect target cells in accordance with the manufacturer's instructions. ISRE/NF-κB/STAT3/SMAD Cignal Reporter lentivirus particles were purchased from QIAGEN. Lentivirus particles expressing Cas9 (Edit-R Lentiviral Cas9 Nuclease) and single-guide (sg)RNA for human CEACAM6 (Edit-R Lentiviral sgRNA, VSGH10148-EG4680, #23) were purchased from

Dharmacon. HEK293 cells were infected with human or cynomolgus CEACAM6-expressing lentivirus. HPAF-II/Cas9/Fluc and HPAF-II/Cas9/CEACAM6-KO/Rluc cells were established by infection with lentivirus expressing Cas9, sgRNA for human CEACAM6, Fluc, and Rluc. LSCs and PSCs expressing Fluc signal reporter along with Rluc were established by infection with Cignal Reporter lentivirus and Rluc-expressing lentivirus. PC-3 cells were infected with Fluc-expressing lentivirus (PC-3/Fluc). HEK293 cells were dually infected with LgBiT-expressing lentivirus and BD1-HiBiT-expressing lentivirus (293/LgBiT/BD1-HiBiT).

## Establishment of Pan02/P-glycoprotein-KO/hCEACAM6 cells

Murine *Abcb1a* and *Abcb1b* genes were simultaneously targeted in Pan02 cells by using the CRISPR/Cas9 ribonucleoprotein method. Recombinant Cas9 proteins (TrueCut Cas9 v2, Invitrogen) and sgRNAs (TrueGuide gRNA, Invitrogen) were obtained from Thermo Fisher. The genomic target sequences for the sgRNAs were as follows: *Abcb1a*: 5′-TAAGTGGGAGCGCCACTCCA-3′, *Abcb1b*: 5′-TCCAAACACCAGCATCAAGA-3′. Pan02 cells were electroporated with the sgRNAs/Cas9 ribonucleoprotein complexes by using a NEPA21 Super Electroporator (Nepa Gene). The electroporated cells were subjected to Rhodamine-123 staining (Sigma-Aldrich) followed by fluorescence-activated cell sorting to isolate rhodamine-positive P-glycoprotein-KO cells. The isolated Pan02 cells were subsequently infected with human CEACAM6-expressing lentivirus.

## Mouse xenograft study

Only female mice were used in this study, considering the effect of fighting between male mice on the evaluation of drug efficacy. Mice were housed in $23 \pm 3\,°C$, a humidity of $55 \pm 15\%$, and a 12 h dark/light cycle. Body weight and tumor size were measured twice a week, and the formula $V = (d^2 \times D)/2$ (where $d$ = minor tumor axis and $D$ = major tumor axis) was used to determine tumor volume (mm$^3$) or relative tumor volume in relation to the initial tumor volume. Humane endpoints (euthanasia required) are 2000 mm$^3$ tumor volume, 20% decrease of body weight from baseline, and significant deterioration of general body condition. In this study, there were no instances of deviation from these provisions. Complete regression (100% reduction of tumor volume) was defined as relative tumor volume 0.01 on log-scale graphs. For the subcutaneous PDAC-PDX model, the tumor tissues were cut into fragments measuring $3 \times 3 \times 3$ mm and inoculated subcutaneously into the right flank of female NOD-SCID mice (Charles River, 5–6 weeks old) by using a trocar needle. When the tumor volume reached approximately 100 mm$^3$, mice were allocated randomly to each group. GEM (200 mg/kg, Eli Lilly), nab-PTX (50 mg/kg, Taiho Pharmaceutical), and ADCs were intravenously injected into the tail vein on day 0. For the mouse syngeneic tumor model, $5 \times 10^6$ Pan02 cells were subcutaneously injected into the right flank of female C57BL/6 mice (Charles River, 4 weeks old). When the tumor volume reached approximately 50 mm$^3$, mice were allocated randomly to each group. Anti-mouse PD-1 antibody (200 μg/head, BE0146, BioXCell) was intraperitoneally injected on day 0 and day 7, and ADC was intravenously injected into the tail vein on day 0. For the liver metastasis model, female NSG mice (NOD.Cg-PrkdcscidIl2rgtm1Wjl, Charles River, 5–6 weeks old) were anesthetized and laparotomized; this was followed by slow injection of $1.0 \times 10^5$ PC-3/Fluc cells in 50 μL PBS into the portal vein. For the orthotopic transplant model, female NSG mice were anesthetized and laparotomized; $1.0 \times 10^6$ PC-3/Fluc cells in 50 μL Matrigel were slowly injected into the pancreas. As postoperative treatments, Baytril (Bayer) and Antisedan (Zenoaq) were given subcutaneously. Bioluminescence imaging of tumor growth in the liver or pancreas was monitored weekly by using an IVIS Spectrum In Vivo Imaging System (PerkinElmer) after intraperitoneal injection of D-luciferin (Promega). Photon flux emitted from the tumor was quantified by using Living Image Software (PerkinElmer, v3.2.0.8156).

When the total photon flux emission reached approximately $3 \times 10^8$ photons s$^{-1}$, mice were allocated randomly to each group. GEM (200 mg/kg) and ADCs were intravenously injected into the tail vein on day 0.

## Organoid growth inhibition (OGI) assay

PDX tumors were extirpated from mice and digested with a Tumor Dissociation Kit and gentleMACS Dissociator (Miltenyi Biotec). Tumor fragments about 50 μm in diameter were separated by repeated low-speed centrifugation and embedded in 25 μL of medium with 10% Growth Factor Reduced-Matrigel (Corning) on 384-well ultra-low-attachment microplates (Corning) at a concentration of 100 to 150 fragments per well. After solidification of the Matrigel for 2 h at 37 °C, 25 μL fresh medium with samples was added to each well, and the plates were further incubated for 5 days. After the 5 days of culture, 25 μL of CellTiter-Glo 3D reagent (Promega) was added to each well and the luminescence was quantified with an Envision microplate reader (PerkinElmer). Depending on the assay or model, the following kinds of culture medium were used: DMEM/F12 supplemented with 10% embryonic stem-cell FBS (Thermo Fisher) or StemPro hESC SFM (Thermo Fisher) for PC-3 and PC-42 models; DMEM/F12 supplemented with 50 ng/mL human epidermal growth factor (Peprotech), 12.5 ng/mL human FGF (fibroblast growth factor) 10 (Peprotech), 10 mM nicotinamide (Sigma-Aldrich), 0.2% bovine serum albumin (Wako), insulin-transferrin-selenium supplement (Thermo Fisher) and 4i cocktail for KYK models.

## Cell growth inhibition (CGI) assay

AsPC-1 and HPAF-II cells were harvested, diluted in RPMI1640 medium containing 10% FBS, and dispensed in 384-well plates at a concentration of 800 cells per well in 25 μL medium. After overnight incubation of the plates, 25 μL fresh medium with samples was added to each well and the plates were further incubated for 5 days. After the 5-day culture, 25 μL of CellTiter-Glo 2.0 reagent (Promega) was added to each well and the luminescence was quantified with an Envision microplate reader (PerkinElmer).

## Myeloid differentiation assay with human HPCs

Human cord blood CD34$^+$ HPCs were cultured in StemSpan SFEM II containing StemSpan CD34$^+$ expansion supplement (STEMCELL Technologies). To differentiate them into MPCs, the HPCs were transferred to StemSpan SFEM II with StemSpan myeloid expansion supplement (STEMCELL Technologies). The medium was changed every 2 or 3 days, and the MPCs were used after 7 days of differentiation.

## Flow cytometric analysis

PDX-derived PDAC cells, HPCs, MPCs, HSAECs, and HEK293 cells expressing human or cynomolgus CEACAM6 were harvested and washed with PBS containing 2 mM EDTA (Invitrogen) and 0.3% bovine serum albumin (WAKO). Before being stained, the cells were pre-incubated with 20 μL of the human Fc receptor binding inhibitor Clear Back (MBL) per $5 \times 10^5$ cells. The cells were then incubated with antibodies against CEACAM6, #84.7 (10 μg/mL) or KOR (D028-3, MBL, 1:100), mouse H-2Kd/H-2Dd (114718, BioLegend, 1:100), or human EpCAM (5447, Cell Signaling Technology, 1:100). Dead cells and small debris were eliminated in FSC/SSC plots, and then alive cells were separated as DAPI negative cells. Human PDAC cells were examined in human EpCAM positive and mouse H-2Kd/H-2Dd negative cells. To evaluate cell-internalization activity, antibody-treated PC-42 cells were resuspended in DMEM/F12 (Invitrogen) containing 10% FBS (HyClone) after cell wash, divided into two plates, and incubated at 4 °C or 37 °C for another 2 h. After incubation with secondary antibodies conjugated with Alexa Fluor (Invitrogen, 1:400), the cells were analyzed with LSRFortessa Flow Cytometer and FlowJo software (BD Biosciences). Internalization activity was quantitated by comparing mean fluorescence intensity of PC-42 cells incubated at 4 °C and 37 °C. For the absolute quantitation of CEACAM6 on cell surface of PDAC cell lines, PDX-derived PDAC cells, MPCs, and HSAECs were analyzed by QIFKIT (Dako) in accordance with the manufacturers' instructions.

## Mass cytometric analysis

Seven days after drug administration, mice were sacrificed and tumors and draining lymph nodes were collected. Tumors were dissociated using a tumor dissociation kit (Miltenyi Biotec) and gentleMACS Dissociator (Miltenyi Biotec), then CD45-positive cells were separated using mouse TIL (CD45) microbeads (Miltenyi Biotec) and OctoMACS Separator (Miltenyi Biotec). Lymph nodes were grinded, washed and filtered, and then cells were collected. TILs were cultured in RPMI 1640 medium containing 10% FBS, 20 ng/mL PMA (Sigma), 500 ng/mL Ionomycin (Sigma), and BD GolgiPlug (BD Biosciences) for 3 h before staining. Cell staining was performed according to the manufacturer's instructions. Briefly, $1.5 \times 10^6$ cells were first incubated with 0.1 μmol/L Cisplatin-198 (Fluidigm) for dead cell exclusion, then blocked with 1/20 diluted anti-mouse CD16/32 (BD Biosciences), stained with extracellular antibodies (Supplementary Table). The cells were fixed with Maxpar Fix I Buffer (Fluidigm), permeabilized with Maxpar Perm-S Buffer (Fluidigm), and stained with intracellular antibodies. Next, they were fixed with 1.6% paraformaldehyde and labeled with Cell-ID intercalator-Ir (Fluidigm). Finally, cells were washed and suspended in Maxpar water, then 0.1X EQ Four Element Calibration Beads (Fluidigm) were added and mixed before acquisition on Helios a CyTOF system (Fluidigm). The measured data were analyzed with Cytobank Premium (Beckman Coulter, v10.1).

## ALI culture of human lung epithelial cells

HPAEpiCs were harvested and seeded at a density of 300,000 cells/well in PneumaCult-Ex Plus media on culture inserts for 24-well plates (Corning) coated with 0.3 mg/mL Cellmatrix Type I-C (Nitta Gelatin). They were cultured for 2 or 3 days. When the cells had reached confluence, the apical medium was removed and the basal medium was replaced with PneumaCult-ALI medium (STEMCELL Technologies). The basal medium was changed every second day for 10 to 14 days until differentiation was well established. Transepithelial electrical resistance was measured to determine the integrity of the epithelial layers by using cellZscope (CellSeed), and wells with a transepithelial electrical resistance ≥400 Ω.cm$^2$ were selected. The cells in the selected wells were cultured for an additional 5 days after ADC addition to the basal medium. Surviving epithelial cells were stained with crystal violet (Sigma-Aldrich), and images were captured with BZ-810 microscope (Keyence). For immunohistochemical staining, the insert membrane was cut out and a formalin-fixed paraffin-embedded (FFPE) block was prepared. Sections (5 μm) were deparaffinized, antigen retrieval was performed, and 3% $H_2O_2$ was used to block endogenous peroxidases. The antigen-retrieval reagents and primary antibodies were, respectively, EGFR (ab52894, Abcam, 1:100): Dako Target Retrieval Solution, pH 9.0 (Dako); HER2 (2165, Cell Signaling Technology, 1:100): immunohistochemical antigen retrieval reagent, pH 8.0 (Enzo); and CEACAM6 (#84.7, 10 μg/mL): Dako Target Retrieval Solution, pH 9.0 (Dako). Detection was performed by using a SignalStain DAB Substrate Kit (Cell Signaling Technology), and hematoxylin was used as a nuclear counterstain. Whole digital slide images were obtained by using an Aperio AT2 slide scanner (Leica Biosystems).

## In vivo distribution analysis of antibody in cynomolgus monkeys

Cynomolgus monkeys (Tian Hu Cambodia Animal Breeding Research Center Ltd., 3–4 years old) were maintained in the institution approved by Accreditation of Laboratory Animal Care International, and animal experiments were conducted in accordance with the guidelines of Shin Nihon Kagaku, Ltd. Monkeys were housed in 26 ± 3 °C, a humidity of

50 ± 20%, and a 12 h dark/light cycle and given a single intravenous injection of 7.5 mg/kg of the following antibodies (female, $n = 1$): HEL3 mAb, CEACAM6 mAb (#84.7), EGFR mAb (cetuximab, Merck), or HER2 mAb (trastuzumab, Chugai). Antibody biodistribution in normal tissues was assessed by immunofluorescence staining 24 h after injection.

### HEK293/LgBiT/BD1-HiBiT assay
HEK293/LgBiT/BD1-HiBiT cells were harvested and seeded in 384-well plates at a concentration of 3000 cells per well in 25 µL DMEM medium containing 10% FBS and Nano-Glo Endurazine Live Cell Substrate (Promega). After 3 h of incubation of the cells, 25 µL of sample dilution was added to each well, and the plates were further incubated. Luminescence was quantified with an Envision microplate reader (PerkinElmer).

### In vitro stromal signaling assay
LSCs expressing Rluc and Fluc signal reporters were used. Before the LSCs were harvested, the culture medium was changed to MSCGM medium (Lonza) and the LSCs were further incubated for 24 h. In the co-culture experiments, $1 × 10^3$ LSCs or the same number of PC-3 cancer cells, or both, were seeded on collagen I-coated 384-well plates (Greiner Bio-one) in 50 µL MSCGM medium per well. The culture medium was replaced with 50 µL of fresh medium with samples 1 day after plating. Three days after treatment, the Fluc signal (reporter activity) and Rluc signal (cell viability) were quantified by using a Dual-Glo Luciferase Assay system (Promega) and an Envision microplate reader (PerkinElmer) in accordance with the manufacturers' instructions.

### In vitro bystander efficacy assay
Target HPAF-II cells (HPAF-II/Cas9/Fluc) and bystander HPAF-II cells (HPAF-II/CEACAM6-KO/Rluc) were harvested and seeded on 384-well plates at four densities in 25 µL medium, with target to bystander ratios of 0:600, 200:400, 400:200, and 600:0. After 3 h of incubation of the wells at 37 °C, 25 µL of fresh medium with samples was added to each well. Five days after treatment, the Fluc signal (target viability) and Rluc signal (bystander viability) were quantified by using a Dual-Glo Luciferase Assay system (Promega) and Envision microplate reader (PerkinElmer) in accordance with the manufacturers' instructions.

### In vitro cytokine secretion assay in co-culture of PDAC cells and CAFs
Mouse CAFs ($1 × 10^5$) or the same number of PC-3 cancer cells, or both, were seeded on a collagen-I-coated 6-well plate (Iwaki) in 2 mL MSCGM per well. The culture medium was replaced with 2 mL of sample dilution 1 day after plating. After 2 days of culture, the culture supernatants were collected and mouse IL-6 and mouse LIF were quantified by using Quantikine ELISA kits (R&D Systems) in accordance with the manufacturers' instructions.

### In vitro tri-culture assay of PDAC cells, CAFs, and PBMCs
PC-3/Fluc cells ($5 × 10^2$) or the same number of CAF/Rluc cells, or both, were seeded on collagen I-coated 384-well plates (Greiner Bio-one) in 25 µL MSCGM medium per well. After overnight incubation of the plates at 37 °C, PBMCs (25 µL of $2.5 × 10^3$ or $1 × 10^4$ cells) with or without anti-human PD-1 antibody (BE0188, BioXCell, 3.3 or 30 µg/mL) or ADCs were added to each well. Seventy-two or ninety-six hours after the treatment, Fluc and Rluc signals were quantified with a Dual-Glo Luciferase Assay system (Promega) and Envision microplate reader (PerkinElmer) in accordance with the manufacturers' instructions. For cytokine secretion assay, 1 mL of $1 × 10^4$ PC-3/Fluc cells or the same number of CAF/Rluc cells, or both, were seeded on collagen I-coated 24-well plates (Iwaki). After overnight incubation of the plates at 37 °C, PBMCs (1 mL of $2.5 × 10^4$ or $5.0 × 10^4$ cells) were added to each well. Culture supernatants were collected 24 or 72 h after the addition of PBMCs, and human IL-2 and human IFN-γ were quantified by using Quantikine ELISA kits (R&D Systems) in accordance with the manufacturer's instructions.

### Western blot analysis
PC-3 cells or mouse CAFs were treated with EBET-1055 or dBET6 at various concentrations. Twenty-four hours after drug treatment, the cells were washed and lysed with Cell Lysis Buffer (Cell Signaling Technology) or 0.25% Triton X buffer containing Halt Protease and Phosphatase Inhibitor Cocktail, EDTA-free (100X) (Thermo Fisher). Collected lysates were electrophoresed and transferred to membranes. The membranes were then incubated with primary antibodies after being blocked with 5% skim milk in Tris-buffered saline and then incubated with secondary antibodies conjugated with horseradish peroxidase. Membrane images were captured with the FUSION Chemiluminescence Imaging System (Vilber-Lourmat). For immunoprecipitation, LSCs were lysed with Pierce IP Lysis Buffer (Thermo Fisher) and fractionated with an EpiQuik Nuclear Extraction Kit II (EpiGentek) to obtain the nuclear fraction. The nuclear fraction was precleared with Pierce Control Agarose Resin (Thermo Fisher), and then incubated overnight at 4 °C with Pierce Protein A/G PLUS-Agarose (Thermo Fisher) conjugated with normal rabbit IgG (2729, Cell Signaling Technology) or BRD4 antibody (A301-985A100, Bethyl, 4 µg/mg lysate). The beads were washed, eluted with sodium dodecyl sulfate sample buffer, and heated at 95 °C for 5 min. The primary antibodies used for western blot were as follows: BRD2 (5848, Cell Signaling Technology, 1:1000), BRD4 (13440, Cell Signaling Technology, 1:1000; or A301-985A100, Bethyl, 1:1000), CDK9 (2316, Cell Signaling Technology, 1:1000), GAPDH (2118, Cell Signaling Technology, 1:1000), pSMAD2^{S465/S467}/pSMAD3^{S423/S425} (8828, Cell Signaling Technology, 1:1000), pSTAT3^{Y705} (9145, Cell Signaling Technology, 1:2000), SMAD2/3 (8685, Cell Signaling Technology, 1:1000), and STAT3 (4904, Cell Signaling Technology, 1:2000).

### Immunofluorescence analysis of organoids and tissues
For organoid staining, tumor fragments were embedded in 50% GFR-Matrigel (Corning) on an eight-well Imaging Chamber (Zell-Kontakt). After 5 days of culture or drug treatment, organoids were fixed with 4% paraformaldehyde, permeabilized with cold methanol, and stained with antibodies against E-cadherin (61081, BD Pharmingen, 1:200; or AF748, R&D Systems, 1:50), Ki-67 (PA5-19462, Thermo, 1:100), collagen I (ab34710, Abcam, 1:100), BRD2 (5848, Cell Signaling Technology, 1:200), BRD4 (ab128874, Abcam, 1:100), fluorescent-dye-labeled rBC2LCN (Fujifilm, 1:200), and DAPI (Dojindo). For tumor staining, PDX tumors were embedded in OCT compound (Sakura) and cryosections were prepared. The sections were fixed with cold methanol and stained with antibodies against E-cadherin (AF748, R&D Systems, 1:50), Ki-67 (PA5-19462, Thermo Fisher, 1:100), and collagen I (ab34710, Abcam, 1:100), fluorescent-dye-labeled rBC2LCN (Fujifilm, 1:200) and DAPI (Dojindo). For lung staining, a normal monkey lung was extirpated and embedded in OCT compound (Sakura), and cryosections were prepared. Sections were fixed with cold methanol and stained with antibodies against CEACAM6 (#84.7, 10 µg/mL), E-cadherin (61081, BD Pharmingen, 1:200), EGFR (cetuximab, Merck, 10 µg/mL), HER2 (2165, Cell Signaling Technology, 1:200), and DAPI (Dojindo). For in vivo distribution analysis of antibodies, monkey lung was extirpated 24 h after injection of the antibodies and embedded in OCT compound (Sakura), and cryosections were prepared. Sections were fixed with cold methanol and stained with antibodies against CEACAM6 (ab134074, Abcam, 1:100), E-cadherin (61081, BD Pharmingen. 1:200), EGFR (ab52894, Abcam, 1:100), HER2 (2165, Cell Signaling Technology, 1:100), human IgG (A80-319A, Bethyl, 1:100) and DAPI (Dojindo). All the images were captured with a TCS SP8 confocal microscope (Leica Biosystems) and analyzed with Leica Application Suite X (Leica Biosystems) and Image J software (National Institutes of Health).

## Immunohistochemical analysis of tissues

For staining of xenograft tumors, 5-µm sections from FFPE tumor samples were deparaffinized, and antigen retrieval was performed by using Dako Target Retrieval Solution, pH 6.0 (Dako). Endogenous peroxidases were blocked with 3% $H_2O_2$. The primary antibodies were as follows: BRD4 (A301-985A100, Bethly, 1:6000), pSTAT3$^{Y705}$ (ab76315, Abcam, 1:100), pSMAD2$^{S465/S467}$ (44–244G, Thermo Fisher, 1:100), pSMAD3$^{S423/S425}$ (ab52903, Abcam, 1:100), collagen I (ab34710, Abcam, 1:50), CD8a (ab217344, Abcam, 1:500), CD4 (ab183685, Abcam, 1:500), and granzyme B (AF1865, R&D Systems, 1:40). For CEACAM6 staining of xenograft tumors, antigen retrieval was performed by using a REAL Target Retrieval Solution, pH 9.0 (Dako). Endogenous peroxidases were blocked with 3% $H_2O_2$, and the sections were incubated with #84.7 CEACAM6 antibody (10 µg/mL). Detection was performed with a SignalStain DAB Substrate Kit (Cell Signaling Technology) or ImmPACT DAB EqV (Vector Laboratories), and hematoxylin was used as a nuclear counterstain. For double staining of PDGFRα and α-SMA of xenograft tumors, antigen retrieval was performed with SignalStain Citrate Unmasking Solution (Cell Signaling Technology) and REAL Target Retrieval Solution, pH 6.0 (Dako). To block endogenous peroxidases, 3% $H_2O_2$ was used. The primary antibodies were PDGFRα (3174, Cell Signaling Technology, 1:200) and α-SMA (A5691, Sigma-Aldrich, 1:100), and the substrates were SignalStain DAB Substrate Kit (Cell Signaling Technology) and Ferangi Blue Chromogen Kit 2 (Biocare Medical). For CEACAM6 staining of clinical PDAC samples, all procedures were performed automatically in BenchMark (Ventana Medical Systems). Sections (3 µm) from FFPE tumor samples were deparaffinized, and antigen retrieval was performed with CC2 solution at 100 °C for 64 min. Endogenous peroxidases were blocked with 3% $H_2O_2$, and the sections were incubated with CEACAM6 antibody (D028-3/KOR, MBL, 1:12000). Detection was performed with an iVIEW DAB Detection Kit (Ventana Medical Systems). Hematoxylin was used as a nuclear counterstain. For quantitative evaluations, whole digital slide images were obtained by using an Aperio AT2 slide scanner (Leica Biosystems). Digital images of tumor tissues were annotated as areas of tumor, stroma, and necrosis. The area stained by each antibody was quantified by using HALO image analysis software (Indica Labs, v2.3.2089.69).

## Public database analysis

For target selection of ADC, we used GEPIA (http://gepia.cancer-pku.cn/index.html), which is an interactive server for analyzing the RNA sequencing expression data of 9736 tumors and 8587 normal samples from the TCGA and GTEx projects by using a standard processing pipeline. A dot plot of CEACAM6 expression and a survival plot based on CEACAM6 expression were drawn by using GEPIA web tools.

## RNA sequence analysis

For bulk RNA sequencing of PC-3 and PC-42 tumors, total RNA was extracted by using a Maxwell RSC simplyRNA Tissue Kit (Promega), and the cDNA library was prepared with an Agilent SureSelect Strand Specific RNA Library Prep Kit (Agilent). Reads (150 bp, paired-end) were sequenced on Illumina NovaSeq (12 Gb per sample). Reads were separated by species of origin by using Human.GRCh38 and Mouse.mm10. For single-cell analysis, PC-3 and PC-42 tumors were dissociated by using a tumor dissociation kit (Miltenyi Biotec). After stromal cell isolation with a Mouse Cell Depletion Kit (Miltenyi Biotec), single-cell suspensions of mouse stromal cells (-5000 cells) were loaded into a 10x Chromium Controller (10x Genomics) with a Single Cell 3′ v2 reagent kit (10x Genomics). The libraries were sequenced on a HiSeq X platform (Illumina), and the sequenced reads were processed by using CellRanger software (10x Genomics, v3.0.2). Stromal data from PC-3 and PC-42 tumors were merged, and clustering by t-SNE was performed with Loupe Browser (10x Genomics, v5.1.0). RNA sequence data of KYK models were obtained from KAN Research Institute. RNA sequence data of TCGA were downloaded from UCSC Xena Data Hubs.

**Molecular subtyping of PDAC.** Cluster analysis of tumor subtypes (classical or basal type), stromal subtypes (normal or activated type), and Myc subtypes (Myc signal low or high type) was conducted as defined in the literature[10,11]. Hierarchical clustering was performed by using Ward's method and Euclidean distance in Strand NGS software (Agilent Technologies, v3.4).

## Gene-set enrichment analysis

Expression data of mouse stromal genes with >1 TPM were extracted from bulk or single-cell RNA sequence data of PC-3 and PC-42 tumors. The pathways that were enriched in ADC-treated tumors or cell clusters were analyzed by using GSEA software (Broad Institute, v4.1.0) and MSigDB (Broad Institute, v5.2).

## Statistical analysis

No statistical method was used to predetermine sample size. Data collection and analysis were not performed with blinding to the conditions of the experiments. Sample sizes, statistical analyses, and *P*-values are indicated in the figure legends. GraphPad Prism software (v9) was used for statistical calculations.

## Reporting summary

Further information on research design is available in the Nature Portfolio Reporting Summary linked to this article.

## Data availability

The RNA sequencing data generated for this study are available at the European Genome-phenome Archive (https://ega-archive.org/) with study IDs EGAS00001007070 and EGAS00001007212. RNA-seq data is not publicly available to maintain patient privacy. Therefore, to access NGS data please request access from our data access committee from the EGA links above. The data access committee will send a consent form by e-mail upon request, and access will be granted immediately upon return by the requestor. The RNA sequencing data of TCGA used in this study is available at the UCSC Xena Data Hubs with dataset ID TCGA.PAAD.sampleMap/HiSeqV2. Synthesis and characterization information of the compounds used in this study is available in patents with WO numbers: WO2024043319, WO2009099741, WO2015095223, WO2019044947, WO2003043583. The remaining data are available within the Article, Supplementary Information or Source data file. Source data are provided with this paper.

## Code availability

Gene expression analyses were performed using open-source or commercially available software, CellRanger software (10x Genomics, v3.0.2), Loupe Browser (10x Genomics, v5.1.0), Strand NGS software (Agilent Technologies, v3.4), GSEA software (Broad Institute, v4.1.0) and MSigDB (Broad Institute, v5.2).

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

## Acknowledgements

We thank K. Mizuno, R. Nakatani, Y. Ida, M. Anzai, and Y. Sakamoto (KAN Research Institute, Inc.) for preparation and characterization of CEACAM6 and HEL3 antibodies; T. Oda, O. Shimomura (University of Tsukuba), H. Seno, and S. Ogawa (University of Kyoto) for providing the PDAC-PDX models; H. Azuma, Y. Ando, and Y. Nakane (Eisai Co., Ltd.) for compound development; K. Sagane and K. Kubara (Eisai Co., Ltd.) for providing plasmids; Y. Nagayama, E. Ota, T. Taniguchi, K. Nakajima, and Y. Seki (Eisai Co., Ltd.) for toxicological advice; T. Mochizuki and Y. Nozaki (Eisai Co., Ltd.) for pharmacokinetic advice; J. Ito (Eisai Co., Ltd.) for bioinformatics advice; H. Ui, Y. Watanabe, M. Shimizu, A. Kayano, W. Ichikawa, M. Isomura, Y. Uemoto, K. Tagami, Y. Kamada, M. Kamoto, M. Kato, and A. Akao (Eisai Co., Ltd.) for chemistry, manufacturing and controls advice; S. Nakamori, K. Morimoto (Osaka National Hospital), and Medmain Inc. for pathological advice; J. Spidel, X. Cheng, A. Vaessen, A. Verdi, E. Albone, K. Furuuchi, and T. Uenaka (Epochal Precision Anti-Cancer Therapeutics, Eisai Inc.) for providing the Trop2 antibody; P. Li (Eisai Co., Ltd.) and Sunplanet Co., Ltd. for supporting our experiments; and K. Sasai, T. Imai, S. Watanabe, K. Kira, A. Yokoi, T. Owa, J. Matsui, K. Nomoto, T. Matsushima, A. Tomonari, Y. Otake, and Y. Tanoue (Eisai Co., Ltd.) for useful discussions.

## Author contributions

Y.N. led and managed the whole project. M.M. led the medicinal chemistry part. Y.N., S.T., H.K., T.A., A. Yamaguchi, M.K., K.T., A.M., T.H., and Y.M. performed the experiments. M.M., A. Yamamoto, K.I., S.I., Y. Yamane., Y. Yabe., H.U. and J.T. synthesized the linker payload and ADC. Y.N. wrote the manuscript.

## Competing interests

All authors are employees of Eisai Co., Ltd. and this study was funded by Eisai Co., Ltd.
