## [Peer Review File · Nature Communications]

Reviewers' Comments:

Reviewer #1:

Remarks to the Author:

The authors and company have invested a significant amount of effort on the screening and development of an anti-CEACAM6 with BET protein degrader payload antibody drug conjugate for the treatment of pancreatic cancer. Overall, this is an extensive amount of data using a rational screening approach for both a unique payload with cancer cell and stromal cell effects (known to be important in pancreatic cancer) and antibody with high target expression and modest healthy tissue binding (where some may be inaccessible due to the target location). The authors developed the novel ADC, tested it in several different PDX models of pancreatic cancer, examined antibody binding in cyno, tested the biological impact on CAFs via bystander effects in vivo, and tested combination therapy with the ADC. This work is informative to the community developing novel ADCs. There are a few areas to be addressed listed below.

Overall comments:

The biggest weakness in the development of this compound is the unknown tolerability of the novel payload. Do the authors have an estimate as to the potential clinical tolerability (e.g. cyno tox data, or related compounds)? Colombo et al. 2022, Cancer Cell showed that ADCs often don't increase the tolerability of the payload. Likewise, Nessler et al. 2021, Trends in Pharm Sci have highlighted how sufficient antibody dosing is needed for adequate tumor uptake even with bystander payloads. Without an estimate for the tolerability of the ADC, the potential for clinical translation remains uncertain. This is not to say that the data are not promising. It is simply uncertain whether these results will translate to the clinic without an estimate of tolerability.

Related to the tolerability, what is the expression level on lung and MPCs (receptors/cell)? This will likely determine if a MTD is target-mediated or not. It appears like expression is higher on the lung than the PC-42 cells, although it may not be as accessible as the authors mention. Lung tox has been an issue for multiple ADCs, however.

What is the linker release mechanism of the ADC? Is it protease cleavable, non-cleavable, etc? Is there a self-immolating portion such that the EBET-1593 is the released compound? This information appears to be missing from the manuscript.

The language of 'potential curability' should be removed from the manuscript. While the data in mice look promising, preclinical studies are premature for discussions on curability, particularly given the daunting challenges associated with pancreatic cancer.

Specific comments:

Page 4: PC-3 is a common prostate cancer cell line, so it may be helpful to explicitly state this is a different (pancreatic) line, since this proprietary model would be less commonly known.

Page 5 and 6: PNU, calicheamicin, and PBD agents have pM potency, and some have bystander effects. Why were the IC50 values thousands of times lower? Are cancer cells dying but fibroblasts surviving? Is there another explanation?

Page 8: What is the expression level of CEACAM6 (receptors/cell)? IHC staining and H-scores are not very informative for quantifying the expression level. For instance, 3+ IHC staining can be 1 million receptors per cell or 6 thousand receptors per cell (e.g. Sharma et al. 2017, Cancer Res).

Page 9: What is the mechanism behind the specificity? Is it related to the affinity? Glycosylation? Have the authors examined the epitope on the antigen?

Page 13: It is hard to say if there's no toxicity enhancement, since body weight is a crude indicator. A more nuanced statement would be more appropriate.

Figure 2 – There is little quantification in these images, making it difficult to determine the level of

differences. Is there any way to quantify these results?

Figure 2d – How is internalization assessed in this figure? While internalization can occur at 37C, the same signal is generated, so it could be surface binding as well.

What is the membrane permeability of the degrader (e.g. PAMPA)? These larger BRo5 compounds are often more challenging to get into/out of cells.

Reviewer #2:

Remarks to the Author:

Nakazawa and colleagues identify and test a novel ADC for the treatment of pancreatic cancer. The manuscript represents a comprehensive preclinical workup of a new therapeutic strategy. The authors identify that a bromodomain and extra-terminal (BET) degrader, termed EBET is effective as a payload for an ADC and then choose CEACAM6 as the target for delivery. The authors develop a novel monoclonal antibody #84.7 specific to CEACAM6. #84.7 appears to have selectivity to CEACAM6 expressed on tumor cells and localize marginally to endogenous CAECAM6 on lung epithelial cells in monkeys. The ADC has impressive single agent activity in PDX models of pancreatic cancer.

There are multiple noteworthy results. 1. ID of EBET as a therapeutic modality using an organoid screen; 2. Demonstration that EBET alters fibroblast activity (reporter assays) in a co-culture system and reduces induction of a stem-like state in tumor cells in vitro; 3. Development of a novel mAb specific for CEACAM6 and characterization of the mAb, including showing limited activity of the ADC in an ALI model and in vivo localization studies in monkeys, which helps validate the concept of targeting CEACAM; 4. Demonstration of in vivo efficacy in PDX models that represent classical or basal PDA; 5. Data suggestive of promising activity of the 84.7-EBET ADC in combination with anti-PD-1 in an engineered model.

The work provides compelling evidence that 84.7-EBET ADC is an attractive therapeutic that is poised for translation to the clinic. The work is original.

The methods are adequately described and the study encompasses a wide breadth of techniques, results of which in general support the overall conclusions of the study.

Comments:

1. A challenge when developing human specific reagents is the limited toxicity of the construct because normal cells in the mouse are not bound by the therapeutic. The authors provide evidence that 84.7-EBET will be well tolerated in humans (ALI model, in vivo localization in monkeys, no weight loss in mice); however the lack of binding of 84.7 to mouse CEACAM6 complicates interpretation of any toxicity. I suggest that the authors 1. report or determine the MTD of EBET alone in mice; 2. Repeat the ALI study using 84.7-EBET instead of the PNU payload; 3. Determine if 84.7-EBET and KOR-EBET show cytotoxic activity on human PBMCs.

2. The authors should provide the following clarifications. 1. How is the ADC produced, what is the linker between IgG and payload; 2. Is that linker used for all the ADCs in the study; 3. How many EBET moieties are on each 84.7 molecule; 4. What is the concentration of EBET (or payload) delivered in the in vitro studies (e.g. Fig 3a,b,c).

3. For all in vivo studies the average size of tumors at the start of therapy should be reported.

4. In Fig 3c is the viability of CAFs reduced by treatment with any of the ADCs?

Suggested edits:

A general edit for clarity is encouraged.

Reviewer #3:

Remarks to the Author:

In this manuscript, Nakazawa, Miyano and colleagues engineer a novel ADC using a novel BET protein degrader (EBET) conjugated to a newly developed antibody to CEACAM6. They test the novel degrader on both PDAC organoids and bystander CAFs, while the CEACAM6 antibody is tested in monkey tissues and used against transplanted organoids in mice. The authors also leverage a human co-culture system that brings organoids, CAFs and PBMCs together. Finally, they test the ADC in tumor transplant models and observe dramatic response when they combine the ADC with PD-1 checkpoint immunotherapy.

The novelty of the manuscript is high as a novel drug and antibody are described. The robustness of the in vitro studies is also high as the authors test a large cohort of patient-derived organoids. The mechanisms of action of the ADC is investigated and seem to indicate that bystander effect on CAFs is important. The in vivo validation, while striking, may be problematic as CEACAM6 is not present in mouse tissues and therefore the results may be confounded. As a result, the conclusions may be overstated.

The manuscript could benefit from a careful review as details appear to be missing and figure descriptions are not always accurate. As presented the manuscript is very confusing.

Major comments:

1. "curability" is a strong term considering that the in vivo models used do not really represent the human disease. Mouse cells do not express the CEACAM6 antigen and therefore the tumor cells expressing CEACAM6 make a perfect target for the ADC.
2. The introduction is brief. No discussion is provided for the history of BET inhibitors in PDAC. It would be helpful to introduce these for the readers that may not be familiar. Also, on pages 16-17 there is a whole paragraph dedicated to introducing chemotherapy regimens in PDAC. This could be in the introduction rather than discussion.
3. CEACAM6 is not present in mouse cells, therefore using this as the target for the ADC would bias the binding to human tumor cells in PDX models. This is not really discussed. How do the authors plan on addressing this issue in future pre-clinical trials?
4. The clustering shown in FIGS1 for activated vs normal stroma is not very convincing. Color legend is missing from the figure or legend. Why are there fewer samples in the MYC signature heatmap?
5. FIGS1b: having the two PDX models on two separate graphs makes comparison of response very challenging especially since the scales of the graphs are different. Please re-graph.
6. The call outs in the text for FigS2 a and b seem to be in the wrong order compared to the figure.
7. "All reporter activities on LSCs were enhanced by LSC co-culture with PC-3 cancer cells (Fig. 1b)" this is not what figure 1b shows. How do the authors make this conclusion?
8. How was this done: "We therefore selected EBET-1055 as a seed compound, optimized it as a payload, and finally obtained the lead payload EBET-1593 (Extended Data Fig. 3b)." There are no details provided in the methods or results.
9. How does 84-EBET affect CAF viability in a mono and co-culture in vitro assay?
10. The single cell data is poorly described and lacks context. As written, I am not sure if anything new was learned. Were the mice treated for the single cell experiment?
11. The triple co-culture experiments were not autologous, can the authors expand on how meaningful the immune reaction is and much of this phenotype would be translatable to patients where checkpoint inhibition often has no impact on tumor cells? Would using a mouse syngeneic co culture system expressing CEACAM6 be more useful since the authors have access to such a model?
12. Can the authors show representative images of the data quantified in figure 6C (this can be added to supplemental figures). In relationship to this experiment, have the authors also checked markers of proliferation and cell death in these mice? Is it possible that the immune reaction observed is due to the overexpression of CEACAM6 on the tumor cells, thus providing a tumor-specific antigen? How would this translate to patients? Finally, it would be beneficial to break down the BRD4+ staining by cell type (tumor vs CAF staining) to demonstrate in this curative model that there is a bystander effect on the CAF BRD4 expression. Similarly staining the tissue with p-STAT3

and a CAF marker would help support the model proposed by the authors.

13. Can the authors discuss why their novel ADC outperforms BRD2/4 inhibitors in PDAC? Would other epigenetic targeted drugs perform better in PDAC if they were conjugated to a tumor specific antibody?

Minor issues:

1. Line numbers would make reviewing easier.
2. In the introduction, second paragraph, the "#84.7" is repeated twice in a row.
3. Please clarify if organoids were directly made from PDX tumors and used immediately for assays or if they were passaged in culture first.

Reviewer #4:

Remarks to the Author:

Authors of this manuscript reported the development of antibody-BET (bromodomain and Extra-Terminal domain) degrader conjugated molecules, represented by 84-EBET, that delivered the payload (BET degrader) to pancreatic cancer cells overexpressing CEACAM6. In addition, the BET degrader payload (such as EBET-1055) was shown to attenuate proinflammatory and pro-survival genes in tumor-associated fibroblasts that support pancreatic tumor growth. Attenuation of the tumor microenvironment by EBET-1055 would be particularly beneficial for pancreatic cancers because of the dense fibrotic environment surrounding pancreatic cancers seen in vivo models and in clinics. As a single agent, 84-EBET dose-dependently suppressed the tumor growth in xenograft mouse models transplanted with four different pancreatic cancer cells expressing high levels of CEACAM6 in vitro. Interestingly, 84-EBET was also effective in mouse xenograft models transplanted with two of four additional pancreatic cancer cells expressing low levels of CEACAM6. 84-EBET as a single agent was shown to be significantly more effective than gemcitabine (current standard therapy for pancreatic cancers) or HEL-EBET (non-targeting payload control) in PC-3 orthotopic transplanted model. Although 84-EBET reduced a set of pro-fibrotic cytokines expression in the tumor and the tumor-associated fibroblasts, 84-EBET synergized more effectively with PD-1 monoclonal antibody in the Pan02/hCECAM6 mouse model without causing toxicity. In summary, the authors first developed a novel class of BET inhibitors that were used to generate BET degrader based on the Proteolysis-targeting chimera (PROTAC) technology. The BET degrader molecules were subsequently coupled to an in-house developed antibody recognizing the pancreatic cancer cells specifically expressed surface marker (CEACAM6) for delivery. By recognizing the BET degrader's utility in suppressing pro-growth inflammatory genes, the authors characterized that 84-EBET suppressed the pro-inflammatory environment at the tumor site via the bystander effect. Finally, 84-EBET can synergize with the immune checkpoint inhibitor (PD1-mAb) to induce a sustained anti-tumor activity in the orthotopic pancreatic cancer cell line mouse model. Overall, this is a comprehensive study that integrates several state-of-the-art drug development technologies to advance the therapeutic development for pancreatic cancers, one of the leading lethal cancers lacking effective therapy. Although the study design was straightforward, I have the following points for the authors to clarify.

1. Chemistry for preparing the new EBET compounds needs to be either provided or referenced.
2. PC-3 name has been widely used for another prostate cancer cell line. I suggested the authors use a different name to avoid confusion. Is this cell line the same as the BxPC4 from the ATCC or a different cell line? Also, are PC-3 and PC-42 PDX cell lines?
3. How 84-EBET was generated from EBET-1593 and the #84.7 antibody was not clearly described. EBET-1593 did not appear to have many functional groups for conjugation to a second antibody. Was is the mechanism of the release of EBET-1593 payload from 84-EBET after internalization? Some descriptions of the generation of 84-EBET are needed.
4. What are the blue and red curves in Fig. 3a? Different compounds?
5. In Fig. 3c, was the non-targeting HEL-EBET expected to be internalized? HEL-EBET appeared to dose-dependent reduction of IL-6.
6. A previous study reported that JQ1 inhibited the growth of a subset of pancreatic cancer cell lines grown in 3-dimensional collagen (Mol. Can. Ther., 2014, 13, 1907). In Extended Fig. 3, JQ1

and dBET6 showed poor activity against PC-3 but EBET-590 was more effective. What are the binding affinities differences between JQ1 and EBET-590 against the BET proteins? Was the thousand-fold difference between JQ1 and EBET-590 due to their differences in the binding affinities?

7. In Extended Fig. 4, the degradation of BRD2 and BRD4 by EBET-1055 was shown. What was the effect of EBET-1055 on BRD3 levels?

8. PC-42 is a basal-like cell line and CEACAM6^{low}. Why #84.7 showed stronger internalization activity in PC-42 (page 9)?

9. In Fig. 4, a payload with either CEACAM6 targeting or non-targeting (HEL) was used in the mouse study. Was EBET-1593 alone effective in some of these cell lines? In relation to point 5, was HEL-EBET internalized or did HEL conjugation prevent the HEL-EBET to be internalized? The little antitumor effects of HEL-EBET implied that it may not be internalized in cells.

10. Any insight on why CEACAM6^{low} PC-3 and KYK-036 were sensitive to the CEACAM6 targeting 84-EBET single agent in Fig. 4?

11. In Fig. 6c BRD4⁺, two groups were found in the combo treatment at day 7. Can the authors check if the significance were **** relative to the vehicle?

12. The data from the combination of 84-EBET and PD-1-mAb was encouraging and promising. Although increased antigen presentation by PROTAC molecules may increase the recognition of the cancer cells by activated T cells (J of immunology, 2021, 207, 493), BET protein inhibition suppressed PD-L1 expression in triple-negative breast cancer cells (Cancer Lett, 2019, 265, 45) and PD-1 expression in T cells (Cell Death & Disease, 2022, 13, 671). In the context of pancreatic cancers, was there any data indicating the impact of the degradation of BET proteins (or BET inhibitors) on the PD-L1/PD-1 expression on the pancreatic cancers or the cell lines used in this study? In addition, cytotoxic T-cell activation is required to sustain the long-lasting efficacy of immune checkpoint inhibitors (PD-1-mAb). Fig. 6 showed that 84-EBET treatment reduced the CD4/8/GzmB⁺ positive area. What are the impacts of the BET degradation by 84-EBET on CD8⁺ T cell activation?

13. In Fig. 6, no tumor volume difference was found between 84-EBET and combo treatment up to day 20. This is also reflected in little changes in the population of CD4 and CD8 positive or GzmB positive cells between 83-EBET and combo at days 3 and 7. Dramatic tumor volume reduction in the combo treatment group however occurred at around day 24 (Fig. 6b) but not in the PD-1-mAB treatment group. Do the authors have any interpretation of the dramatic effect of the combo treatment on the tumor volumes at a later timepoint? Was it mediated by the reduction of caf to allow T cell infiltration? Why the CD4 and CD8 staining in the tumor area were not done after day 20 that may help explain the treatment outcome?

14. Do the authors have the data regarding the half-life of 84-EBET in mice?

Minor:

1. Page 4, should be "complex interaction between cancer cells and stromal.."

2. Page 7, improve the immunosuppressive environment will favor the tumor growth. Is this what the authors meant?

3. Page 30, Real-time BD1 degradation assay. Should be "HEK293/LgBiT/BD1-HiBiT".

4. What are the blue curves and dots in Extended Fig. 3c, 3d? Different compounds in the library?

Reviewer #1 – ADC early development (Remarks to the Author):

The authors and company have invested a significant amount of effort on the screening and development of an anti-CEACAM6 with BET protein degrader payload antibody drug conjugate for the treatment of pancreatic cancer. Overall, this is an extensive amount of data using a rational screening approach for both a unique payload with cancer cell and stromal cell effects (known to be important in pancreatic cancer) and antibody with high target expression and modest healthy tissue binding (where some may be inaccessible due to the target location). The authors developed the novel ADC, tested it in several different PDX models of pancreatic cancer, examined antibody binding in cyno, tested the biological impact on CAFs via bystander effects in vivo, and tested combination therapy with the ADC. This work is informative to the community developing novel ADCs. There are a few areas to be addressed listed below.

Overall comments:

1. The biggest weakness in the development of this compound is the unknown tolerability of the novel payload. Do the authors have an estimate as to the potential clinical tolerability (e.g. cyno tox data, or related compounds)? Colombo et al. 2022, Cancer Cell showed that ADCs often don't increase the tolerability of the payload. Likewise, Nessler et al. 2021, Trends in Pharm Sci have highlighted how sufficient antibody dosing is needed for adequate tumor uptake even with bystander payloads. Without an estimate for the tolerability of the ADC, the potential for clinical translation remains uncertain. This is not to say that the data are not promising. It is simply uncertain whether these results will translate to the clinic without an estimate of tolerability.

➤ **This manuscript is a report of lead CEACAM6-EBET-ADC. After evaluation of the lead ADC, final optimization of linker payload has been conducted especially focusing assays with human lung epithelium, hematopoietic progenitor cells and myeloid progenitor cells to improve the tolerability. The final ADC candidate did not show any effect on their viability at 3.3 nM but showed lethal effect in PDAC cells at 0.1 nM. The ADC is currently in tolerability testing with dogs and monkeys for clinical trial. Due to our development policies, we are unable to show the structure of the final candidate here but added the data of in vitro efficacy and toxicity and the explanation (Extended Data Fig. 13a-c, lines 2-6 on page 18).**

2. Related to the tolerability, what is the expression level on lung and MPCs (receptors/cell)? This will likely determine if a MTD is target-mediated or not. It appears like expression is higher on the lung than the PC-42 cells, although it may not be as accessible as the authors mention. Lung tox has been an issue for multiple ADCs, however.

➤ **Human MPCs and lung epithelial cells (HSAEC) showed relatively lower expression of CEACAM6 in vitro (13,210/MPC and 384,704/HSAEC), while PDAC cells showed higher expression ranging from 393,904/cell to 1,753,688/cell. As noted above, we believe we have achieved a certain amount of therapeutic window considering in vitro efficacy and tolerability assays of the final candidate. We added the data quantifying the number of CEACAM6 molecules on the cell membrane (Extended Data Fig. 6b).**

3. What is the linker release mechanism of the ADC? Is it protease cleavable, non-cleavable, etc? Is there a self-immolating portion such that the EBET-1593 is the released compound? This information appears to be missing from the manuscript.

➤ **In this manuscript, we are using cathepsin-B-cleavable GGFG linker. EBET-1593 is released after linker cleavage and self-immolation of aminomethylene moiety. We added the explanation in the manuscript (lines 18-19 on page 7, lines 3-5 on page 26).**

4. The language of 'potential curability' should be removed from the manuscript. While the data in mice look promising, preclinical studies are premature for discussions on curability, particularly given the daunting challenges associated with pancreatic cancer.

➤ **We changed the words to "potential efficacy" (line 11 on page 2, line 1 on page 19).**

Specific comments:

1. Page 4: PC-3 is a common prostate cancer cell line, so it may be helpful to explicitly state this is a different (pancreatic) line, since this proprietary model would be less commonly known.

➤ **We added the explanation (line 11 on page 5).**

2. Page 5 and 6: PNU, calicheamicin, and PBD agents have pM potency, and some have bystander effects. Why were the IC50 values thousands of times lower? Are cancer cells dying but fibroblasts surviving? Is there another explanation?

➤ **Some drugs show lower efficacy in 3D culture systems than in 2D culture, and this is often due to the drug's ability to penetrate the tissue. Our bystander assay of ADCs showed EBET compounds had higher cell permeability compared to DNA-binding reagents. PROTACs induce protein degradation through a catalytic process and are constantly recycled, while DNA-binding reagents require covalent or strong binding to DNA for their efficacy. This might contribute the higher efficacy of EBET in organoid models.**

3. Page 8: What is the expression level of CEACAM6 (receptors/cell)? IHC staining and H-scores are not very informative for quantifying the expression level. For instance, 3+ IHC staining can be 1 million receptors per cell or 6 thousand receptors per cell (e.g. Sharma et al. 2017, Cancer Res).

➤ **We added the data quantifying the number of CEACAM6 molecules on the cell membrane (Extended Data Fig. 6b). Establishment of scoring method of CEACAM6 expression is ongoing for clinical studies.**

4. Page 9: What is the mechanism behind the specificity? Is it related to the affinity? Glycosylation? Have the authors examined the epitope on the antigen?

➤ **We already identified the epitope of our mAb using non-glycosylated CEACAM6. We are continuing our research assuming that different glycosylation pattern on CEACAM6 can alter the recognition by #84.7. The data will be reported separately.**

5. Page 13: It is hard to say if there's no toxicity enhancement, since body weight is a crude indicator. A more nuanced statement would be more appropriate.

➤ **We changed the words to say that there was no additional body weight loss by the combination (lines 7-8 on page 15).**

6. Figure 2 – There is little quantification in these images, making it difficult to determine the level of differences. Is there any way to quantify these results?

➤ **Which images? In case of monkey biodistribution assay (Fig 2f), due to the limitations of antibody application in IF staining, we used unfixed fresh-frozen lungs. This made it difficult to unbiasedly prepare a sufficient number of tissue images for quantitation, and only representative images are presented here. Also, at the time of experiment, it was not possible to prepare a sufficient number of monkeys for quantitation.**

7. Figure 2d – How is internalization assessed in this figure? While internalization can occur at 37C, the same signal is generated, so it could be surface binding as well.

- **Antibody-treated PDAC cells were resuspended in FBS-containing medium after cell wash, divided into two plates, and incubated at 4 °C or 37 °C for another 2 h. After incubation with secondary antibodies conjugated with fluorescent dye, the cells were analyzed by flow cytometry. Internalization activity was quantitated by comparing mean fluorescence intensity of cells incubated at 4 °C and 37 °C. We modified the figure to see the difference clearly and added the explanation to the manuscript (Fig. 2d, lines 17 on page 10 to 1 on page 11, lines 13-17 on page 32).**

8. What is the membrane permeability of the degrader (e.g. PAMPA)? These larger BRo5 compounds are often more challenging to get into/out of cells.

- **EBET compounds have low solubility, and their membrane permeability cannot be measured with PAMPA. However, as mentioned above, the bystander and organoid assays suggest high membrane permeability of EBET compounds.**

Reviewer #2 – PDAC therapy, fibroblasts (Remarks to the Author):

Nakazawa and colleagues identify and test a novel ADC for the treatment of pancreatic cancer. The manuscript represents a comprehensive preclinical workup of a new therapeutic strategy. The authors identify that a bromodomain and extra-terminal (BET) degrader, termed EBET is effective as a payload for an ADC and then choose CEACAM6 as the target for delivery. The authors develop a novel monoclonal antibody #84.7 specific to CEACAM6. #84.7 appears to have selectivity to CEACAM6 expressed on tumor cells and localize marginally to endogenous CAECAM6 on lung epithelial cells in monkeys. The ADC has impressive single agent activity in PDX models of pancreatic cancer.

There are multiple noteworthy results. 1. ID of EBET as a therapeutic modality using an organoid screen; 2. Demonstration that EBET alters fibroblast activity (reporter assays) in a co-culture system and reduces induction of a stem-like state in tumor cells in vitro; 3. Development of a novel mAb specific for CEACAM6 and characterization of the mAb, including showing limited activity of the ADC in an ALI model and in vivo localization studies in monkeys, which helps validate the concept of targeting CEACAM; 4. Demonstration of in vivo efficacy in PDX models that represent classical or basal PDA; 5. Data suggestive of promising activity of the 84.7-EBET ADC in combination with anti-PD-1 in an engineered model. The work provides compelling evidence that 84.7-EBET ADC is an attractive therapeutic that is poised for translation to the clinic. The work is original.

The methods are adequately described and the study encompasses a wide breadth of techniques, results of which in general support the overall conclusions of the study.

Comments:

1. A challenge when developing human specific reagents is the limited toxicity of the construct because normal cells in the mouse are not bound by the therapeutic. The authors provide evidence that 84.7-EBET will be well tolerated in humans (ALI model, in vivo localization in monkeys, no weight loss in mice); however the lack of binding of 84.7 to mouse CEACAM6 complicates interpretation of any toxicity. I suggest that the authors 1. report or determine the MTD of EBET alone in mice; 2. Repeat the ALI study using 84.7-EBET instead of the PNU payload; 3. Determine if 84.7-EBET and KOR-EBET show cytotoxic activity on human PBMCs.

- **About suggestion 1. We determined MTD of final candidate of EBET in mice and rats (1 mg/kg). This dose is equivalent to 35 mg/kg of ADC when calculated by the amount of loaded payload.**
- **About suggestion 2 and 3. This manuscript is a report of lead CEACAM6-EBET-ADC. After evaluation of the lead ADC, final optimization of linker payload has been conducted especially focusing culture assays with human lung epithelium, hematopoietic progenitor cells and myeloid progenitor cells to improve the tolerability. The final ADC candidate did not show any effect on their viability at 3.3 nM but showed lethal effect in PDAC cells at 0.1 nM. Therefore, we believe we have achieved a certain amount of therapeutic window of the final candidate. The ADC is currently in tolerability testing with dogs and monkeys for clinical trial. Due to our development policies, we are unable to show the structure of the final candidate here but added the data of in vitro efficacy and toxicity and the explanation (Extended Data Fig. 13a-c, lines 2-6 on page 18).**

2. The authors should provide the following clarifications. 1. How is the ADC produced, what is the linker between IgG and payload; 2. Is that linker used for all the ADCs in the study; 3. How many EBET moieties are on each 84.7 molecule; 4. What is the concentration of EBET (or payload) delivered in the in vitro studies (e.g. Fig 3a,b,c).

- **In this manuscript, we are using cathepsin-B-cleavable GGFG linker. EBET-1593 is released after linker cleavage and self-immolation of aminomethylene moiety. Drug antibody ratio is 4, then the concentration of EBET in the in vitro**

studies are 4 times higher than the stated concentration of ADC in the figures. We added the explanation in the manuscript (**lines 18-19 on page 7, lines 3-5 on page 26, line 2 on page 12**).

3. For all in vivo studies the average size of tumors at the start of therapy should be reported.

➤ **We added the info in Figure legends (Fig 4a, 4b, 6a, 6b and Extended Data Fig 1b).**

4. In Fig 3c is the viability of CAFs reduced by treatment with any of the ADCs?

➤ **84-EBET treatment did not affect CAF viability under the conditions of this co-culture experiment. We added the data (Extended Data Fig. 6e).**

Suggested edits:

A general edit for clarity is encouraged.

Reviewer #3 – PDAC therapy, organoids & PDX (Remarks to the Author):

In this manuscript, Nakazawa, Miyano and colleagues engineer a novel ADC using a novel BET protein degrader (EBET) conjugated to a newly developed antibody to CEACAM6. They test the novel degrader on both PDAC organoids and bystander CAFs, while the CEACAM6 antibody is tested in monkey tissues and used against transplanted organoids in mice. The authors also leverage a human co-culture system that brings organoids, CAFs and PBMCs together. Finally, they test the ADC in tumor transplant models and observe dramatic response when they combine the ADC with PD-1 checkpoint immunotherapy.

The novelty of the manuscript is high as a novel drug and antibody are described. The robustness of the in vitro studies is also high as the authors test a large cohort of patient-derived organoids. The mechanisms of action of the ADC is investigated and seem to indicate that bystander effect on CAFs is important. The in vivo validation, while striking, may be problematic as CEACAM6 is not present in mouse tissues and therefore the results may be confounded. As a result, the conclusions may be overstated.

The manuscript could benefit from a careful review as details appear to be missing and figure descriptions are not always accurate. As presented the manuscript is very confusing.

Major comments:

1. “curability” is a strong term considering that the in vivo models used do not really represent the human disease. Mouse cells do not express the CEACAM6 antigen and therefore the tumor cells expressing CEACAM6 make a perfect target for the ADC.

➤ **We changed the words to “potential efficacy” (line 11 on page 2, line 1 on page 19).**

2. The introduction is brief. No discussion is provided for the history of BET inhibitors in PDAC. It would be helpful to introduce these for the readers that may not be familiar. Also, on pages 16-17 there is a whole paragraph dedicated to introducing chemotherapy regimens in PDAC. This could be in the introduction rather than discussion.

➤ **We added the explanation about chemotherapy regimens for PDAC and history of BET inhibitors in Introduction (lines 13 on page 3 to 7 on page 4).**

3. CEACAM6 is not present in mouse cells, therefore using this as the target for the ADC would bias the binding to human tumor cells in PDX models. This is not really discussed. How do the authors plan on addressing this issue in future pre-clinical trials?

➤ **This manuscript is a report of lead CEACAM6-EBET-ADC. After evaluation of the lead ADC, final optimization of linker payload has been conducted especially focusing culture assays with human lung epithelium, hematopoietic progenitor cells and myeloid progenitor cells to improve the tolerability. The final ADC candidate did not show any effect on their viability at 3.3 nM but showed lethal effect in PDAC cells at 0.1 nM. Therefore, we believe we have achieved a certain amount of therapeutic window of the final candidate. The ADC is currently in tolerability testing with dogs and monkeys for clinical trial. Due to our development policies, we are unable to show the structure of the final candidate here but added the data of in vitro efficacy and toxicity and the explanation (Extended Data Fig. 13a-c, lines 2-6 on page 18).**

4. The clustering shown in FIGS1 for activated vs normal stroma is not very convincing. Color legend is missing from the figure or legend. Why are there fewer samples in the MYC signature heatmap?

- **We excluded the stromal clustering from the manuscript (Extended Data Fig. 1a).**
 - **We added the explanation about z-score coloring and gene sets in the figure. In the MYC signature clustering, the number of samples on the vertical axis is not changed, but the number of genes on the horizontal axis is smaller than in other clustering (Extended Data Fig. 1a).**
5. FIGS1b: having the two PDX models on two separate graphs makes comparison of response very challenging especially since the scales of the graphs are different. Please re-graph.
- **We modified the graphs. Here, we focused whether GEM treatment induced tumor shrinkage or not. We added the explanation in the manuscript (Extended Data Fig. 1b, lines 15-16 on page 5).**
6. The call outs in the text for FigS2 a and b seem to be in the wrong order compared to the figure.
- **They are in the correct order. First, we talked about IF result of tumors, then about IF result of organoids. We added the explanation in the manuscript (lines 4-9 on page 6).**
7. "All reporter activities on LSCs were enhanced by LSC co-culture with PC-3 cancer cells (Fig. 1b)" this is not what figure 1b shows. How do the authors make this conclusion?
- **The values on the y-axis are relative to the monoculture, and a value greater than 1 indicates that the reporter activity is enhanced by the co-culture. We added basal values without drug treatment in the manuscript (line 11 on page 7).**
8. How was this done: "We therefore selected EBET-1055 as a seed compound, optimized it as a payload, and finally obtained the lead payload EBET-1593 (Extended Data Fig. 3b)." There are no details provided in the methods or results.
- **We provided the outline of screening flow in Methods (lines 9-15 on page 25).**
9. How does 84-EBET affect CAF viability in a mono and co-culture in vitro assay?
- **84-EBET treatment did not affect CAF viability under the conditions of this co-culture experiment. We added the data (Extended Data Fig. 6e).**
10. The single cell data is poorly described and lacks context. As written, I am not sure if anything new was learned. Were the mice treated for the single cell experiment?
- **Because there are many reports on CAF markers, we used single cell RNAseq data to determine which markers would be appropriate in our models. Specifically, we obtained GSEA data among clusters defined by known CAF markers and determined that the markers were valid based on the enrichment of signalling pathways. We used non-treated PDX tumors for the single cell analysis. We added the explanation in the manuscript (lines 15-16 on page 13, lines 4-5 on page 14).**
11. The triple co-culture experiments were not autologous, can the authors expand on how meaningful the immune reaction is and much of this phenotype would be translatable to patients where checkpoint inhibition often has no impact on tumor cells? Would using a mouse syngeneic co culture system expressing CEACAM6 be more useful since the authors have access to such a model?
- **Tumor cell killing in this tri-culture is a reaction to xenoantigens. Considering the recent reports about neoantigen in PDAC (for example, <https://www.nature.com/articles/s41586-023-06063-y>), T cells capable of responding to**

tumor cells actually exist in PDAC patients, but they are restrained by the immunosuppressive tumor microenvironment. Although this tri-culture induces a reaction to xenoantigens, immunosuppression by CAFs was clearly observed (Extended Data Fig 10a). We have also tried to establish syngeneic tri-culture system, but there was no sufficient tumor cell killing using T cells isolated from the lymph nodes and spleen from tumor-bearing mice.

12. Can the authors show representative images of the data quantified in figure 6C (this can be added to supplemental figures). In relationship to this experiment, have the authors also checked markers of proliferation and cell death in these mice? Is it possible that the immune reaction observed is due to the overexpression of CEACAM6 on the tumor cells, thus providing a tumor-specific antigen? How would this translate to patients? Finally, it would be beneficial to break down the BRD4+ staining by cell type (tumor vs CAF staining) to demonstrate in this curative model that there is a bystander effect on the CAF BRD4 expression. Similarly staining the tissue with p-STAT3 and a CAF marker would help support the model proposed by the authors.

- **We added representative images (Extended Data Fig 11).**
- **We have not stained markers of proliferation and cell death. In vitro studies showed that BRD4 degradation is linked to cancer cell death.**
- **Since overexpression of CEACAM6 did not delay tumor engraftment and growth, we think it is unlikely that CEACAM6 induces strong anti-tumor immune response by serving as a tumor antigen. As mentioned above, tumor-reactive T cells are present in PDAC patients, and the immunosuppressive microenvironment is likely to be responsible for impeding the immune response. According to our panel analysis of mouse syngeneic models, Pan02 model showed moderate infiltration of T cells and higher infiltration of inflammatory CAFs and did not show significant response to PD-1-mAb treatment. Therefore, we think this model could be an appropriate model for clinical PDAC.**
- **In Pan02 model, we could not distinguish tumor cells and stromal cells by our HALO system. We stained and quantitated CAF markers, PDGFR α and α SMA, and found that 84-EBET-ADC treatment decreased both markers (Fig 6c, upper right graph).**

13. Can the authors discuss why their novel ADC outperforms BRD2/4 inhibitors in PDAC? Would other epigenetic targeted drugs perform better in PDAC if they were conjugated to a tumor specific antibody?

- **Our compound library contains BET inhibitors and other epigenetic modulators. But only our BET degrader showed lethal effect in organoid screening and broad inhibitory effect in stromal signalling screening. We believe that these characteristics of BET degrader were decisive for its high efficacy in PDAC.**

Minor issues:

1. Line numbers would make reviewing easier.

- **We added line numbers in the manuscript.**

2. In the introduction, second paragraph, the “#84.7” is repeated twice in a row.

- **The first “#84.7” is noted as the clone name of the antibody, and the second “#84.7” is noted to identify the antibody used for the ADC in another sentence. If you have a better wording, please let us know.**

3. Please clarify if organoids were directly made from PDX tumors and used immediately for assays or if they were passaged in culture first.

- **As we mentioned in Methods, organoids were directly made from tumor fragments of PDX tumors and used immediately without passage for organoid growth inhibition assays (lines 1 to 6 on page 31).**

Reviewer #4 - BET protein degraders, non-human primate (Remarks to the Author):

Authors of this manuscript reported the development of antibody-BET (bromodomain and Extra-Terminal domain) degrader conjugated molecules, represented by 84-EBET, that delivered the payload (BET degrader) to pancreatic cancer cells overexpressing CEACAM6. In addition, the BET degrader payload (such as EBET-1055) was shown to attenuate proinflammatory and pro-survival genes in tumor-associated fibroblasts that support pancreatic tumor growth. Attenuation of the tumor microenvironment by EBET-1055 would be particularly beneficial for pancreatic cancers because of the dense fibrotic environment surrounding pancreatic cancers seen in vivo models and in clinics. As a single agent, 84-EBET dose-dependently suppressed the tumor growth in xenograft mouse models transplanted with four different pancreatic cancer cells expressing high levels of CEACAM6 in vitro. Interestingly, 84-EBET was also effective in mouse xenograft models transplanted with two of four additional pancreatic cancer cells expressing low levels of CEACAM6. 84-EBET as a single agent was shown to be significantly more effective than gemcitabine (current standard therapy for pancreatic cancers) or HEL-EBET (non-targeting payload control) in PC-3 orthotopic transplanted model. Although 84-EBET reduced a set of pro-fibrotic cytokines expression in the tumor and the tumor-associated fibroblasts, 84-EBET synergized more effectively with PD-1 monoclonal antibody in the Pan02/hCECAM6 mouse model without causing toxicity. In summary, the authors first developed a novel class of BET inhibitors that were used to generate BET degrader based on the Proteolysis-targeting chimera (PROTAC) technology. The BET degrader molecules were subsequently coupled to an in-house developed antibody recognizing the pancreatic cancer cells specifically expressed surface marker (CEACAM6) for delivery. By recognizing the BET degrader's utility in suppressing pro-growth inflammatory genes, the authors characterized that 84-EBET suppressed the pro-inflammatory environment at the tumor site via the bystander effect. Finally, 84-EBET can synergize with the immune checkpoint inhibitor (PD1-mAb) to induce a sustained anti-tumor activity in the orthotopic pancreatic cancer cell line mouse model. Overall, this is a comprehensive study that integrates several state-of-the-art drug development technologies to advance the therapeutic development for pancreatic cancers, one of the leading lethal cancers lacking effective therapy. Although the study design was straightforward, I have the following points for the authors to clarify.

Major:

1. Chemistry for preparing the new EBET compounds needs to be either provided or referenced.
 - **Since the details of the synthesis are very lengthy, we would like to cite our patent in this manuscript. The patent will be published in February 2024, so it would be available for reference by the time this paper is published. I added the patent application number in Methods (line 8-9 on page 25).**
2. PC-3 name has been widely used for another prostate cancer cell line. I suggested the authors use a different name to avoid confusion. Is this cell line the same as the BxPC4 from the ATCC or a different cell line? Also, are PC-3 and PC-42 PDX cell lines?
 - **Since PC-3 model has already been published under this name by the distributor, the name cannot be changed. We stated this is not a prostate cancer model in the manuscript (line 11 on page 5).**
3. How 84-EBET was generated from EBET-1593 and the #84.7 antibody was not clearly described. EBET-15935 did not appear to have many functional groups for conjugation to a second antibody. Was is the mechanism of the release of EBET-1593 payload from 84-EBET after internalization? Some descriptions of the generation of 84-EBET are needed.

➤ **We are using cathepsin-B-cleavable GGFG linker. GGFG linker is linked to the hydroxyl group of EBET-1593 via aminomethylene moiety. EBET-1593 is released after linker cleavage and self-immolation of aminomethylene moiety. We added the explanation in the manuscript (lines 18-19 on page 7, lines 3-5 on page 26).**

4. What are the blue and red curves in Fig. 3a? Different compounds?

➤ **Different kinds of models. Color legends are located at the bottom of Fig 3a.**

5. In Fig. 3c, was the non-targeting HEL-EBET expected to be internalized? HEL-EBET appeared to dose-dependent reduction of IL-6.

➤ **According to the recent report (<https://aacrjournals.org/cancerdiscovery/article/11/7/1808/666549/>), both cancer cells and CAFs in PDAC show macropinocytosis that mediates non-selective fluid-phase uptake. That would be one of the reasons why non-targeting HEL-EBET affect the cytokine secretion from CAFs.**

6. A previous study reported that JQ1 inhibited the growth of a subset of pancreatic cancer cell lines grown in 3-dimensional collagen (Mol. Can. Ther., 2014, 13, 1907). In Extended Fig. 3, JQ1 and dBET6 showed poor activity against PC-3 but EBET-590 was more effective. What are the binding affinities differences between JQ1 and EBET-590 against the BET proteins? Was the thousand-fold difference between JQ1 and EBET-590 due to their differences in the binding affinities?

➤ **We observed a several hundred-fold difference in the binding affinity to BRD2/4 between EBET-590 and JQ1 with bromodomain TR-FRET assays. Consistent with the difference, EBET-590 showed much lower IC50s than JQ1 in 2D CGI assay using several cancer cell lines and organoid growth inhibition assays.**

7. In Extended Fig. 4, the degradation of BRD2 and BRD4 by EBET-1055 was shown. What was the effect of EBET-1055 on BRD3 levels?

➤ **Yes. We found that EBET degraded BRD2, BRD3 and BRD4 in bromodomain-containing family proteins by proteome analysis.**

8. PC-42 is a basal-like cell line and CEACAM6low. Why #84.7 showed stronger internalization activity in PC-42 (page 9)?

➤ **Here, we showed #84.7 had stronger internalization activity compared to commercially available antibody. We don't know exactly why, but there are reports that different Her2 antibodies possess different internalization activities. We modified the figure to see the difference clearly and added the explanation to the manuscript (Fig. 2d, lines 17 on page 10 to 1 on page 11).**

9. In Fig. 4, a payload with either CEACAM6 targeting or non-targeting (HEL) was used in the mouse study. Was EBET-1593 alone effective in some of these cell lines? In relation to point 5, was HEL-EBET internalized or did HEL conjugation prevent the HEL-EBET to be internalized? The little antitumor effects of HEL-EBET implied that it may not be internalized in cells.

➤ **Free payload, EBET-1593 alone, was not effective in mouse xenograft models because of its low tolerability. That's why we are using ADC for specific delivery of EBET.**

➤ **We think that nonspecific uptake of HEL-EBET into tumors occurred to some extent, e.g., via macropinocytosis as mentioned above, but was insufficient to exert an anti-tumor effect in vivo. Decrease of BRD4 proteins in cancer and stromal area in HEL-EBET-treated groups also supported the non-specific uptake (Fig 5b, upper graphs)**

10. Any insight on why CEACAM6^{low} PC-3 and KYK-036 were sensitive to the CEACAM6 targeting 84-EBET single agent in Fig. 4?

- **The classification here is based on IHC scoring, but according to the quantitation data by flow cytometry (Fig 4c), PC-3 and KYK-036 models belong to the medium expression group. On the other hand, CEACAM6^{low}/basal models showed the lowest expression by flow cytometry. We think this is one of the reasons for the difference in drug efficacy. Establishment of more precise scoring method of CEACAM6 expression is ongoing for clinical studies.**

11. In Fig. 6c BRD4⁺, two groups were found in the combo treatment at day 7. Can the authors check if the significance were **** relative to the vehicle?

- **This is correct. There was significant difference between vehicle and combo groups with <0.0001 P-value.**

12. The data from the combination of 84-EBET and PD-1-mAb was encouraging and promising. Although increased antigen presentation by PROTAC molecules may increase the recognition of the cancer cells by activated T cells (J of immunology, 2021, 207, 493), BET protein inhibition suppressed PD-L1 expression in triple-negative breast cancer cells (Cancer Lett, 2019, 265, 45) and PD-1 expression in T cells (Cell Death & Disease, 2022, 13, 671). In the context of pancreatic cancers, was there any data indicating the impact of the degradation of BET proteins (or BET inhibitors) on the PD-L1/PD-1 expression on the pancreatic cancers or the cell lines used in this study? In addition, cytotoxic T-cell activation is required to sustain the long-lasting efficacy of immune checkpoint inhibitors (PD-1-mAb). Fig. 6 showed that 84-EBET treatment reduced the CD4/8/GzmB⁺ positive area. What are the impacts of the BET degradation by 84-EBET on CD8⁺ T cell activation?

- **In vivo and vitro transcriptome data showed that 84-EBET reduced PD-L1 expression in PDAC cells to some extent, but the basal expression level of PD-L1 is very low in our PDAC models.**
- **The role of BRD4 in T cell activation remains controversial. Some reports suggest that BRD4 is important in T cell activation and JQ1 inhibits that (<https://www.frontiersin.org/articles/10.3389/fimmu.2021.728082/full>, <https://rupress.org/jem/article/218/8/e20202512/212191/>), and other reports suggest that JQ1 promotes T cell activation (<https://www.nature.com/articles/s41419-022-05123-x>, [https://www.cell.com/molecular-therapy-family/molecular-therapy/fulltext/S1525-0016\(21\)00301-4](https://www.cell.com/molecular-therapy-family/molecular-therapy/fulltext/S1525-0016(21)00301-4)). Although BRD4 protein has recovered to some extent in combination group on day 7 (Fig 6c), activated T cells increased on day 7 and the combined effect was observed much later than day 7. We think it is unlikely that the direct effect of EBET on T cells contribute to the combined effect. We explained our hypothesis in the next session.**

13. In Fig. 6, no tumor volume difference was found between 84-EBET and combo treatment up to day 20. This is also reflected in little changes in the population of CD4 and CD8 positive or GzmB positive cells between 83-EBET and combo at days 3 and 7. Dramatic tumor volume reduction in the combo treatment group however occurred at around day 24 (Fig. 6b) but not in the PD-1-mAB treatment group. Do the authors have any interpretation of the dramatic effect of the combo treatment on the tumor volumes at a later timepoint? Was it mediated by the reduction of caf to allow T cell infiltration? Why the CD4 and CD8 staining in the tumor area were not done after day 20 that may help explain the treatment outcome?

- **Tumors were too small to conduct MOA analysis after day 7. Immune profiling by mass cytometry on day 7 revealed that 84-EBET decreased immunosuppressive cells in tumor, myeloid-derived suppressor cells (MDSCs) and tumor-associated macrophages (TAMs), and increased effector cells in tumor, T cells and NK cells. In tumor-draining lymph nodes, mature DCs and primed T cells increased in the combination group only, suggesting**

enhanced antigen presentation in tumor. Taken together, our hypothesis is that there are two phases of effects on the immune system. In the first phase, CAF modification by 84-EBET treatment inhibits inflammatory response and immunosuppressive cell infiltration, attracts and activates effector cells, and exerts an antitumor effect. In the next phase, cancer cells die, antigen uptake by dendritic cells is enhanced, and activated DCs migrate to their lymph nodes to promote further T cell activation. Since this second phase is observed only on day 7 in the combination group, we believe that there was a difference in the anti-tumor effect after this time point. We added the data and explanation in the manuscript (**Extended Data Fig. 12, lines 19 on page 15 to 8 on page 16**)

14. Do the authors have the data regarding the half-life of 84-EBET in mice?

➤ **It's about 4 days.**

Minor:

1. Page 4, should be "complex interaction between cancer cells and stromal.."

➤ **We modified the words (line 6 on page 5).**

2. Page 7, improve the immunosuppressive environment will favor the tumor growth. Is this what the authors meant?

➤ **STAT3 knockout improve immunosuppressive microenvironment and induces anti-tumor immunity. We modified the words (line 18 on page 8).**

3. Page 30, Real-time BD1 degradation assay. Should be "HEK293/LgBiT/BD1-HiBiT".

➤ **We modified the words (line 1 on page 35).**

4. What are the blue curves and dots in Extended Fig. 3c, 3d? Different compounds in the library?

➤ **The blue curves and dots are EBET-derivatives and reference compounds (line 5 on page 49).**

Reviewers' Comments:

Reviewer #1:

Remarks to the Author:

The authors have sufficiently addressed my concerns in the revision. This is an exciting piece of work with the potential to both directly kill PDAC cells while altering the tumor microenvironment. The results from the cyno tox studies should shed more light on the clinical translatability of the results.

Reviewer #2:

Remarks to the Author:

The authors have adequately addressed the concerns of the prior review, I have no further queries.

Reviewer #3:

Remarks to the Author:

The authors addressed my questions. Their overall revisions significantly improved the manuscript.

Reviewer #4:

Remarks to the Author:

The issues raised by this reviewer have been properly addressed.